

# A century of waiting: description of a new *Epictia* Gray, 1845 (Serpentes: Leptotyphlopidae) based on specimens housed for more than 100 years in the collection of the Natural History Museum Vienna (NMW)

Claudia Koch[1,*], Angele Martins[2,3,*] and Silke Schweiger[4]

[1] Department of Herpetology, Zoologisches Forschungsmuseum Alexander Koenig (ZFMK), Bonn, Germany
[2] Departamento de Ciências Fisiológicas, Instituto de Ciências Biológicas, Campus Darcy Ribeiro, Universidade de Brasília, Brasília, Brazil
[3] Departamento de Vertebrados, Museu Nacional do Rio de Janeiro (UFRJ), Rio de Janeiro, Brazil
[4] 1st Zoological Department, Herpetological Collection, Natural History Museum Vienna (NMW), Vienna, Austria

[*] These authors contributed equally to this work.

Corresponding author
Claudia Koch, c.koch@zfmk.de

## ABSTRACT

We describe a new species of *Epictia* based on eight specimens from Nicaragua collected and housed in the collection of the Natural History Museum Vienna for more than a century. The species differs from the congeners by the combination of external morphological characters: midtail scale rows 10; supralabials two, anterior one large and in broad contact with supraocular; infralabials four; subcaudals 14–19; middorsal scale rows 250–267; supraocular scales present; frontal scale distinct; striped dorsal color pattern with more or less triangular dark blotches on each scale; small white blotch in anterior part of dorsal surface of rostral present in five out of six specimens (two further specimens are lacking their heads); terminal spine and adjacent scales white. Eidonomic species separation from other *Epictia* spp. is also supported by a few qualitative and quantitative differences in vertebrae count and morphology. The new species is putatively assigned to the *Epictia phenops* species group based on external morphological characters and distribution.

## INTRODUCTION

The fossorial threadsnakes of the family Leptotyphlopidae represent about 140 currently recognized species (*Uetz, Freed & Hošek, 2019*) that occur along sub-Saharan Africa and nearby islands, the Arabian Peninsula, in southwest Asia (Leptotyphlopinae and Rhinoleptini) and in the New World (Americas and Antilles) (*Adalsteinsson et al., 2009*). Despite their wide typical Gondwanan distribution, members of this family still account

for one of the least known terrestrial vertebrates (*Adalsteinsson et al., 2009*). This is mostly due to their secretive habits, which make them rarely encountered in the field, except for a few locally abundant species (*McDiarmid, Campbell & Touré, 1999*; *Curcio, Zaher & Rodrigues, 2002*; *Passos, Caramaschi & Pinto, 2005*). Additionally, the systematics of this group is very controversial mostly due to their relatively conserved external morphology (*Passos, Caramaschi & Pinto, 2005*; *Passos, Caramaschi & Pinto, 2006*; *Pinto & Curcio, 2011*; *Martins, Passos & Pinto, 2018*). The genus *Epictia* is the most speciose amongst the subfamily Epictinae, with about 43 species currently recognized (*Uetz, Freed & Hošek, 2019*). Although subjected to a few systematic studies in recent years (e.g., *McCranie & Hedges, 2016*; *Wallach, 2016*) members of this genus still account for several taxonomical issues, which are far from being satisfactorily resolved. Additionally, several species have been described in the past 10 years (e.g., *Arredondo & Zaher, 2010*; *Koch, Venegas & Böhme, 2015*; *Koch, Santa Cruz & Cárdenas, 2016*; *Wallach, 2016*) reinforcing the need for systematic studies of the group in order to evaluate the presence of new and undescribed taxa in the Neotropical region.

While reviewing specimens of the family Leptotyphlopidae from the herpetological collection of the Natural History Museum Vienna, Austria (NMW), the first author came across a series of eight specimens (NMW 15446:1–8) of threadsnakes, that were assigned by Steindachner, a former curator of the collection, to *Epictia albifrons* (*Wagler, 1824*). Steindachner himself donated these specimens to the museum in 1907 and the locality was stated as ''Corinto''. Unfortunately the original description of *E. albifrons* is rather poor, the holotype (ZSM 1348/0) was destroyed during the Second World War and according to *Da Cunha & Do Nascimento (1993)*, *Wallach, Williams & Boundy (2014)*, *Pinto, Franco & Hoogmoed (2018)* the locality data (''in adjacentibus Urbis Para'' = in the proximity of Pará, Brazil) could be erroneous. Furthermore, the species has been confused with *E. goudotii* (*Duméril & Bibron, 1844*) in some earlier literature (*McCranie & Hedges, 2016*). Although a molecular analysis by *Adalsteinsson et al. (2009)* demonstrated that *E. albifrons* and *Epictia goudotii* represent two genetically distinct species, the validity of *E. albifrons* is still under debate (*McCranie & Hedges, 2016*). It is recognized by some authors as a valid species (e.g., *Peters & Orejas-Miranda, 1970*; *Wallach, Williams & Boundy, 2014*; *Natera Mumaw, Esqueda González & Castelaín Fernández, 2015*; *Wallach, 2016*; *Murphy, Rutherford & Jowers, 2016*), while other authors consider *E. albifrons* as a nomen dubium and place it in the synonymy of *Epictia tenella Klauber, 1939* (e.g., *Wilson & Hahn, 1973*; *Franco & Pinto, 2009*). In contrast some authors consider *E. tenella* as a junior synonym of *E. albifrons* (e.g., *Thomas, 1965*; *Hoogmoed & Gruber, 1983*). *Natera Mumaw, Esqueda González & Castelaín Fernández (2015)* designated a neotype for *E. albifrons* (MCZ R-2885) from the vicinity of ''Pará, Brasil'' which was rejected by *Wallach (2016)* who designate the topotype BYU 11490 from the vicinity of Belém, Pará State, Brazil, as the neotype of *E. albifrons*. According to *Wallach (2016) E. albifrons* is restricted to the vicinity of the topotype series in northeastern Brazil, while *E. tenella* is a relatively widespread species in cis-Andean South America. According to his (and former) diagnosis the main difference between both species is the presence (*E. tenella*) or absence (*E. albifrons*) of a contact between supraoculars and the anterior supralabial. *Pinto, Franco & Hoogmoed (2018)* state

that the neotype proposed by *Natera Mumaw, Esqueda González & Castelaín Fernández (2015)* is valid, whereas *Wallach*'s (*2016*) designation of a neotype was an act against the ICZN (1999) code. They further disagree with *Wallach (2016)* regarding the absence of a supralabial-supraocular contact in *E. albifrons* and consider this contact to be present in this species as proposed by *Natera Mumaw, Esqueda González & Castelaín Fernández (2015)*. Whether *E. albifrons* and *E. tenella* represent two distinct or the same species is not conclusively clarified. We herein treat both species as being valid. However, after the analysis of both external and internal morphological characters, we verified that the specimens donated by Steindachner do neither represent *E. albifrons* nor *E. tenella*, but pertain to a new *Epictia* species (see data in 'Results'). Apart from the unequal distribution the new species differs from both species (sensu *Wallach, 2016* and *Natera Mumaw, Esqueda González & Castelaín Fernández, 2015*) by having a higher number of middorsal scale rows (see 'Comparison').

## MATERIALS & METHODS

We compared the new species with all congeners in the genus *Epictia* that are currently recognized by at least some authors as valid species (see comments in *Koch, Santa Cruz & Cárdenas, 2016*). Therefore, we examined 392 specimens (see Appendix) representing 56 species of South American and Mesoamerican Leptotyphlopidae from the following collections: American Museum of Natural History, New York, USA (AMNH), Natural History Museum, London, UK (BMNH), Centro de Ornitología y Biodiversidad, Lima, Peru (CORBIDI), Coleção Herpetológica da Universidade de Brasília, Brasilia, Brazil (CHUNB), Field Museum of Natural History, Chicago, USA (FMNH), Fundación Miguel Lillo, San Miguel de Tucumán, Argentina (FML), Instituto de Bio y Geociencias del Noroeste Argentino, Rosario de Lerma, Argentina (IBIGEO), Instituto Butantan, Sao Paulo, Brazil (IBSP), Museum für Naturkunde, Berlin, Germany (ZMB), Natural History Museum of Los Angeles, Los Angeles, USA (LACM), Laboratório de Zoologia de Vertebrados, Universidade Federal de Ouro Preto, Oure Preto, Brazil (LZV), Museum für Tierkunde in Dresden, Dresden, Germany (MTKD), Museum of Comparative Zoology, Cambridge, USA (MCZ), Muséum National d'Histoire Naturelle, Paris, France (MNHN), Museo de Historia Natural, Universidad Nacional Mayor de San Marcos, Lima, Peru (MUSM), Museu Nacional, Universidade Federal do Rio de Janeiro, Rio de Janeiro, Brazil (MNRJ), Museu de Zoologia da Universidade de São Paulo, Sao Paulo, Brazil (MZUSP), Museu Paraense Emílio Goeldi, Belém, Brazil (MPEG), Museo de Historia Natural de la Universidad Nacional de San Agustín, Arequipa, Peru (MUSA), Natural History Museum, University of Kansas, Lawrence, USA (KU), Oklahoma Museum of Natural History, Norman, USA (OMNH), Natural History Museum Vienna, Vienna, Austria (NMW), Museo de Zoología, Pontificia Universidad Católica del Ecuador, Quito, Ecuador (QCAZ), Senckenberg Museum Frankfurt, Frankfurt, Germany (SMF), San Diego Museum of Natural History, San Diego, USA (SDMNH), National Museum of Natural History, Washington, USA (USNM), Coleção Zoológica da Universidade Federal do Mato Grosso, Cuiabá, Brazil (UFMT), Illinois Natural History Survey, Champaign, USA (UIMNH),

Zoologische Staatssammlung München, Munich, Germany (ZSM), and Zoologisches Forschungsmuseum Alexander Koenig, Bonn, Germany (ZFMK). In addition, we reviewed data on all known species in the genus *Epictia* from the following literature sources (original species descriptions and others): *Wagler (1824)*, *Schlegel (1839)*, *Duméril & Bibron (1844)*, *Peters (1857)*, *Jan (1861)*, *Cope (1875)*, *Werner (1901)*, *Oliver (1937)*, *Klauber (1939)*, *Taylor (1940)*, *Schmidt & Walker (1943)*, *Smith (1943)*, *Smith & Laufe (1945)*, *Legler (1959)*, *Orejas-Miranda (1961)*, *Orejas-Miranda (1964)*, *Orejas-Miranda (1969)*, *Freiberg & Orejas-Miranda (1968)*, *Peters & Orejas-Miranda (1970)*, *Orejas-Miranda & Zug (1974)*, *Hoogmoed (1977)*, *Zug (1977)*, *Laurent (1984)*, *Lancini & Kornacker (1989)*, *Villa (1990)*, *Vanzolini (1996)*, *Lehr et al. (2002)*, *Duellman (2005)*, *Kretzschmar (2006)*, *Börschig (2007)*, *Boundy & Wallach (2008)*, *Arredondo & Zaher (2010)*, *Pinto et al. (2010)*, *McCranie (2011)*, *Francisco, Pinto & Fernandes (2012)*, *Francisco, Pinto & Fernandes (2018)*, *Esqueda González et al. (2015)*, *Koch, Venegas & Böhme (2015)*, *Koch, Santa Cruz & Cárdenas (2016)*, *Natera Mumaw, Esqueda González & Castelaín Fernández (2015)*, *Martins (2016)*, *McCranie & Hedges (2016)*, *Murphy, Rutherford & Jowers (2016)* and *Wallach (2016)*. Data on other South American Leptotyphlopidae were obtained from: *Pinto & Curcio (2011)*, *Pinto & Fernandes (2012)*, *Pinto & Fernandes (2017)* and *Salazar-Valenzuela et al. (2015)*.

Measurements were taken with a digital caliper to the nearest 0.1 mm or with a ruler to the nearest 1.0 mm. The following abbreviations were used: SVL = snout-vent length, TAL = tail length, TL = total length, HW = head width at largest area (level of the parietal scale), HH = head height at highest point, HL = head length from tip of snout to posterior level of skull, (when feeling it tapering to the neck), ED = eye diameter, RES = relative eye size (eye height/ocular shield height), MB = midbody diameter, MT = midtail diameter, MDS = middorsal scale rows from the rostral scale to the terminal spine, V = number of ventral scales in longitudinal row from mental to cloacal shield, D = number of scales around the body counted at three different points along the body (1. at a head's length behind the head, 2. at midbody, 3. at a head's length before the cloaca), SC = number of subcaudal scales counted in longitudinal row from cloaca to tip of tail, TS = number of midtail scale rows counted transversely across the middle of the tail, SL = number of supralabials, IF = number of infralabials, PCV = precloacal vertebrae, CAV = caudal vertebrae, CLV = cloacal vertebrae.

Terminology for cephalic plates, scale features, and measurements follows *Wallach (2003)*, *Broadley & Wallach (2007)*, *Arredondo & Zaher (2010)* and *Francisco, Pinto & Fernandes (2012)*.

For obtaining information on the number of precloacal and caudal vertebrae and the position of the pelvic girdle, whole specimens were X-rayed in 2D outside of ethanol with a Faxitron X-ray LX60 at ZFMK. For details of the skull and lower jaw, cervical vertebrae and pelvic girdle, the head, midbody and cloacal region of each specimen were X-rayed in 3D by the use of a high-resolution micro-CT scanner (Bruker SkyScan 1173) at ZFMK. Therefore bodies of specimens were placed in a tube filled with ethanol and only the regions of interest were sticking out of the ethanol and were CT-scanned (head and cloacal region were scanned together at the same time). Specimens were CT-scanned in 180° degrees at

rotation steps of 0.3° or 0.4° degrees with a tube voltage of 40 kV and a tube current of 200 µA without the use of a filter at an image resolution of 8.4 µm. Scan duration was between 25 min (rotation steps of 0.4°) and 42 min (rotation steps of 0.3°) with an exposure time of 359 ms. The CT-dataset was reconstructed using N-Recon software (Bruker MicroCT) and rendered in three dimensions through the aid of CTVox for Windows 64 bits version 2.6 (Bruker MicroCT). We have also used osteological data from *Martins (2016)* and (A Martins, 2016, unpublished data), specimens cited in the APPENDIX—Additional specimens examined)  to provide additional species comparisons. Anatomical terminology follows *Romer (1956)*, *List (1966)* and *Holman (2000)* [atlas and axis], *List (1966)* [pelvic girdle], *Rieppel (1979)* [parabasisphenoid complex], *Kley (2006)* and *Martins (2016)* [suspensorium], *Cundall & Irish (2008)*, *Rieppel, Kley & Maisano (2009)*, *Martins (2016)* and (A Martins, 2016, unpublished data) [skull], *Curcio (2003)* and *Rieppel, Kley & Maisano (2009)* [skull foramina].

The electronic version of this article in Portable Document Format (PDF) will represent a published work according to the International Commission on Zoological Nomenclature (ICZN), and hence the new names contained in the electronic version are effectively published under that Code from the electronic edition alone. This published work and the nomenclatural acts it contains have been registered in ZooBank, the online registration system for the ICZN. The ZooBank LSIDs (Life Science Identifiers) can be resolved and the associated information viewed through any standard web browser by appending the LSID to the prefix http://zoobank.org/. The LSID for this publication is: urn:lsid:zoobank.org:pub:C2392EA5-9957-45BF-AE8C-0998E342F90B. The online version of this work is archived and available from the following digital repositories: PeerJ, PubMed Central and CLOCKSS.

# RESULTS

## *Epictia rioignis* sp. nov. (Figs. 1–15, Table 1)
urn:lsid:zoobank.org:act:4C7CFC14-FDC7-4B41-B7CF-7CCB8F0E8649

### Holotype
NMW 15446:6, from Corinto, presumably Nicaragua (12°29′N, 87°11′W, see Discussion for further details) donated by Steindachner in 1907.

### Paratypes (7)
NMW 15446:1–5, NMW 15446:7–8 from the type locality, donated by Steindachner in 1907.

### Diagnosis
*Epictia rioignis* sp. nov. can be distinguished from all congeners by the following combination of characters: (1) midbody scale rows 14; (2) midtail scale rows 10; (3) supralabials two, anterior one large and in broad contact with supraocular; (4) infralabials four; (5) subcaudals 14–19; (6) middorsal scale rows 250–267; (7) total number of precloacal vertebrae 231–248; (8) supraocular scales present; (9) frontal scale distinct, not fused with

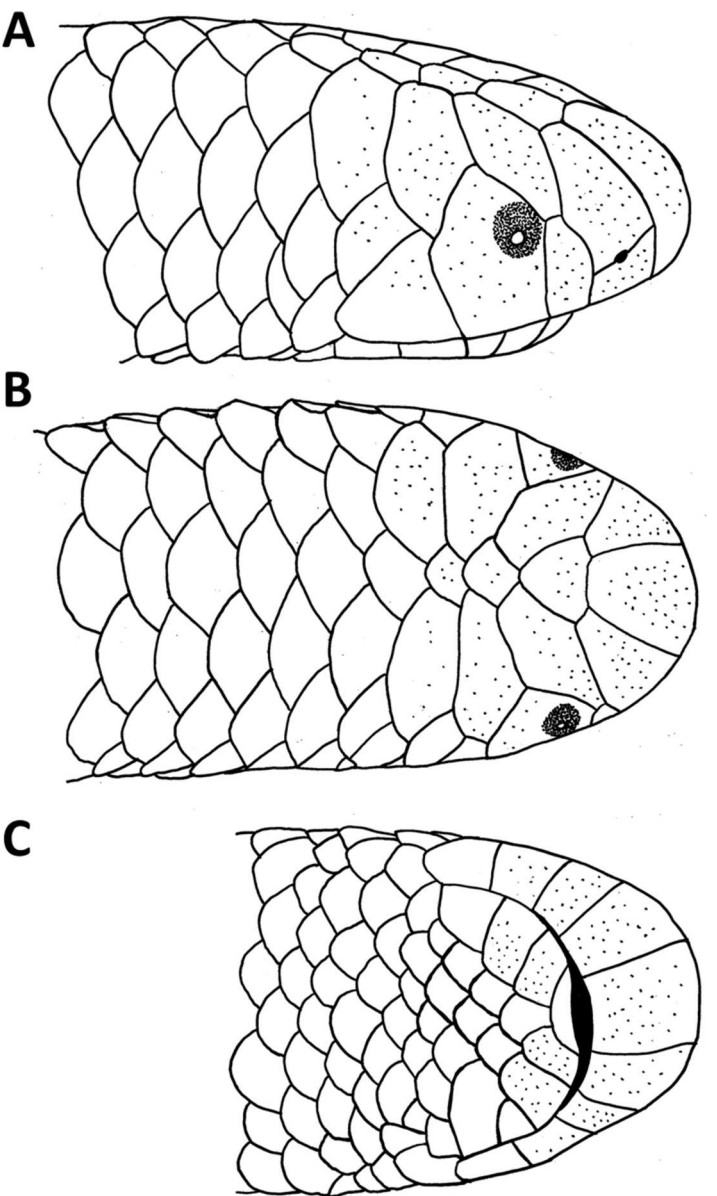

**Figure 1** **Drawings of the head of the holotype of *Epictia rioignis* sp. nov. (NMW 15446:6).** (A) Lateral, (B) dorsal and (C) ventral views.

rostral; (10) striped dorsal color pattern with more or less triangular dark blotches on each scale; (11) upper half of eyes visible in dorsal view; (12) some caudals in posterior part of tail are fused in 50% of the specimens; (13) small white blotch in anterior part of dorsal surface of rostral present in about 83% of the specimens; (14) terminal spine and adjacent scales white.

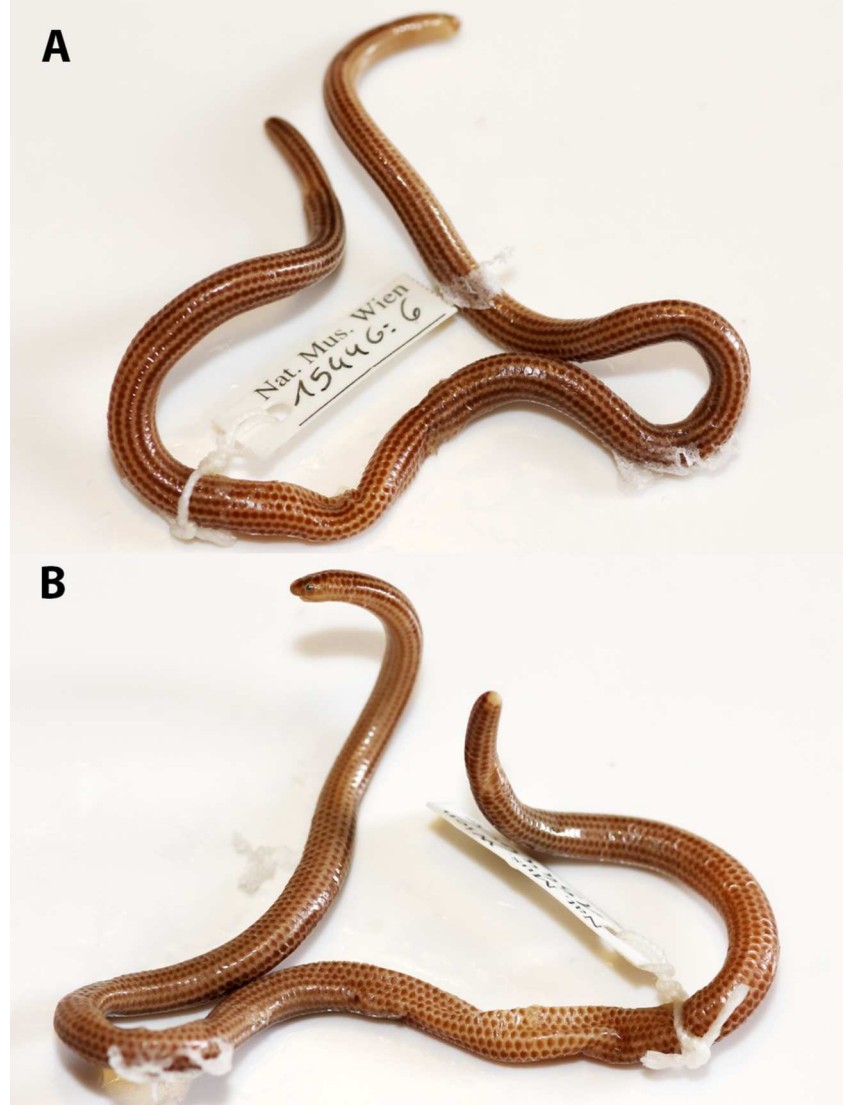

**Figure 2** **Holotype of *Epictia rioignis*. sp. nov. (NMW 15446:6).** (A) Dorsal and (B) ventral or lateral views.

## Comparisons

The new species differs from *Epictia albipuncta, Epictia striatula, Epictia unicolor* and *Epictia weyrauchi* by having 10 midtail scale rows [vs. 12]. The number of 14 midbody scale rows distinguishes the species from *E. undecimstriata* [16]. By the presence of an unfused frontal and rostral scale it is differentiated from *Epictia ater* (including *Epictia nasalis*), *Epictia bakewelli* and *Epictia schneideri*. By having the anterior supralabial in broad contact with the supraocular the new species can be distinguished from *E. albipuncta, Epictia amazonica, E. ater, Epictia australis, E. bakewelli, Epictia borapeliotes, Epictia clinorostris, Epictia collaris, Epictia columbi, Epictia diaplocia, Epictia fallax, Epictia goudotii, Epictia magnamaculata, Epictia martinezi, Epictia melanura, Epictia munoai, Epictia pauldwyeri,*

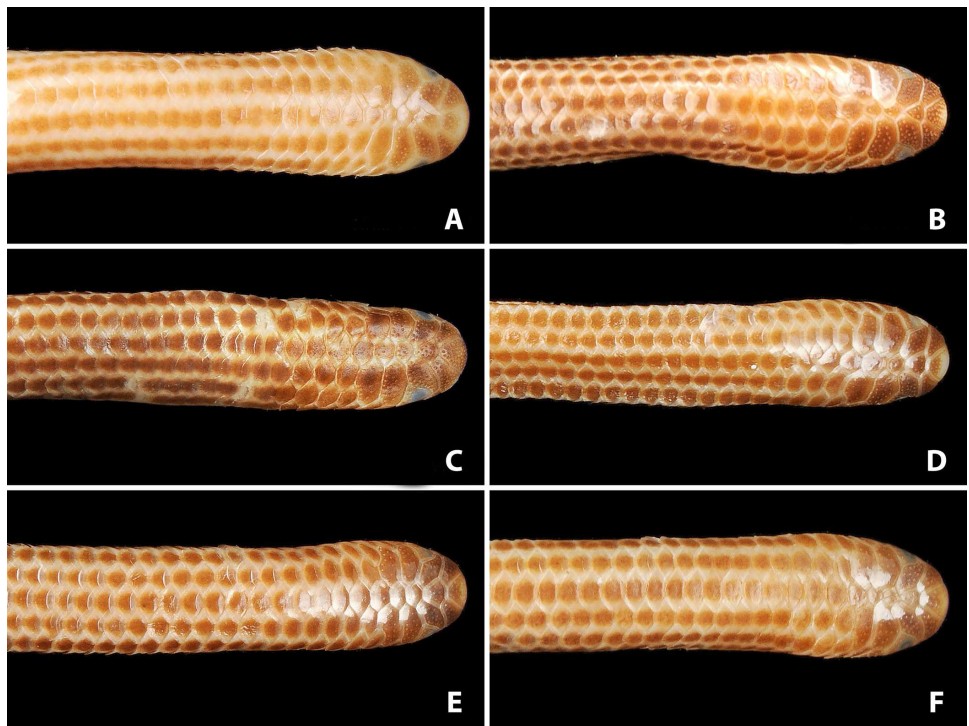

**Figure 3** **Comparison of dorsal views of the heads of holotype (NMW 15446:6, A) and paratypes of *Epictia rioignis* sp. nov. (B–F).** (A) Holotype (NMW 15446:6). (B) Paratype (NMW 15446:2). (C) Paratype (NMW 15446:4). (D) Paratype (NMW 15446:5). (E) Paratype (NMW 15446:7). (F) Paratype (NMW 15446:8). Photo credits: Alice Schumacher & Josef Muhsil.

*Epictia peruviana, Epictia phenops, Epictia resetari, E. schneideri, Epictia signata, Epictia subcrotilla, Epictia vellardi, Epictia vindumi,* and *Epictia wynni* [vs. anterior supralabial and supraocular separated by supranasal-ocular contact]. The number of 250–267 middorsal scale rows distinguishes *Epictia rioignis* sp. nov. from *E. albifrons* [206–218 sensu *Wallach, 2016*; 242 sensu Natera Mumaw, Esqueda González & Castelaín Fernández, 2015], *Epictia alfredschmidti* [267–279], *E. amazonica* [208–245], *Epictia antoniogarciai* [195–208], *E. collaris* [155–166], *E. diaplocia* [205–233], *Epictia hobartsmithi* [191–208], *E. melanura* [395–396], *E. munoai* [184–226], *E. pauldwyeri* [202–226], *E. peruviana* [185–199], *E. subcrotilla* [318–333], *Epictia tenella* [215–233 sensu *Wallach, 2016* ], *Epictia tricolor* [276–310], *E. unicolor* [246], *Epictia vanwallachi* [188], *Epictia venegasi* [211–221], and *Epictia vonmayi* [196–205]. The number of 14–19 subcaudal scales differentiates this species from *E. columbi* [22–25], *E. munoai* [10–14], *E. nasalis* [21], and *E. pauldwyeri* [10–14]. By the presence of four infralabials [vs. three] *Epictia rioignis* sp. nov. differs from *E. australis, E. borapeliotes, E. collaris, E. munoai,* and *E. wynni*. The presence of a light blotch on the dorsal part of the rostral further differentiates the new species from *E. columbi, Epictia rufidorsa, E. vanwallachi,* and *E. weyrauchi*, and the presence of a whitish terminal spine distinguishes it from *E. columbi, E. melanura, Epictia melanoterma, E. rufidorsa,* and *Epictia septemlineata*. By lacking a tricolor pattern (reddish-brown, black, yellow) it differs from

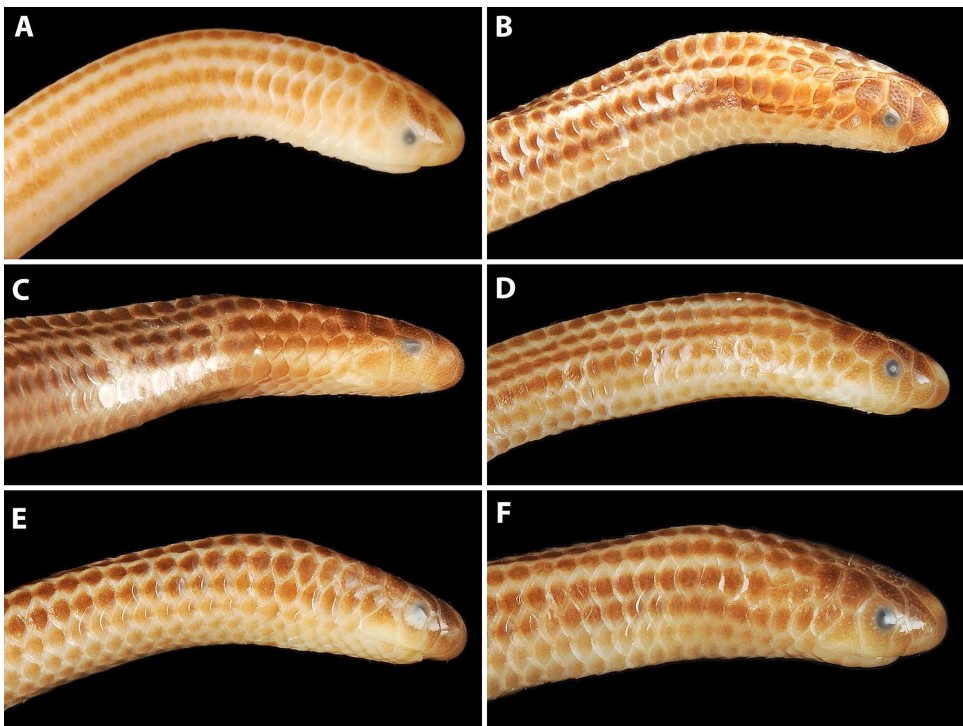

**Figure 4** Comparison of lateral views of the heads of holotype (NMW 15446:6, A) and paratypes of *Epictia rioignis* sp. nov. (B–F). (A) Holotype (NMW 15446:6). (B) Paratype (NMW 15446:2). (C) Paratype (NMW 15446:4). (D) Paratype (NMW 15446:5). (E) Paratype (NMW 15446:7). (F) Paratype (NMW 15446:8). Photo credits: Alice Schumacher & Josef Muhsil.

*E. alfredschmidti, Epictia rubrolineata, Epictia teaguei,* and *E. tricolor.* By lacking a preoral groove in the ventral rostral it differs from *E. columbi.* The presence of a distinct striped dorsal color pattern with more or less triangular dark blotches on each scale distinguishes the new species from *E. amazonica* [uniformly black coloration, without any trace of stripes], *E. ater,* and *E. columbi* [both species appear uniformly dark, pale outline of the scales is only visible upon closer examination]. From *Epictia tesselata* which is only known from Lima (Peru) and surroundings, the new species differs by having a very small light blotch on the rostral [light spot on the rostral and lower portion of the nasals] and a darker ventral coloration. It differs from *E. ater* and *E. phenops* by presenting unfused neural arches of the atlas [vs. fused]. The number of 231–248 trunk vertebrae distinguishes it from *E. magnamaculata* [199], *E. munoai* [207], *E. phenops* [213–246], *E. tenella* [190–204], and *E. tricolor* [282].

## Description of holotype (Figs. 1 and 2, and upper left picture of Figs. 3–9)

A large specimen with TL of 211 mm; TAL of 10.3 mm; MB of 4.4 mm; MT of 3.1 mm; TL/TAL of 20.5; TL/MB of 48; TAL/MT of 3.3; HW of 2.8 mm; HL of 3.6 mm; HH of 2.3 mm; ED of 0.4 mm; RES of 0.3. Head subcylindrical, slightly dorsoventrally compressed, hardly distinguishable from neck; body cylindrical; not tapered cranially or caudally. Snout

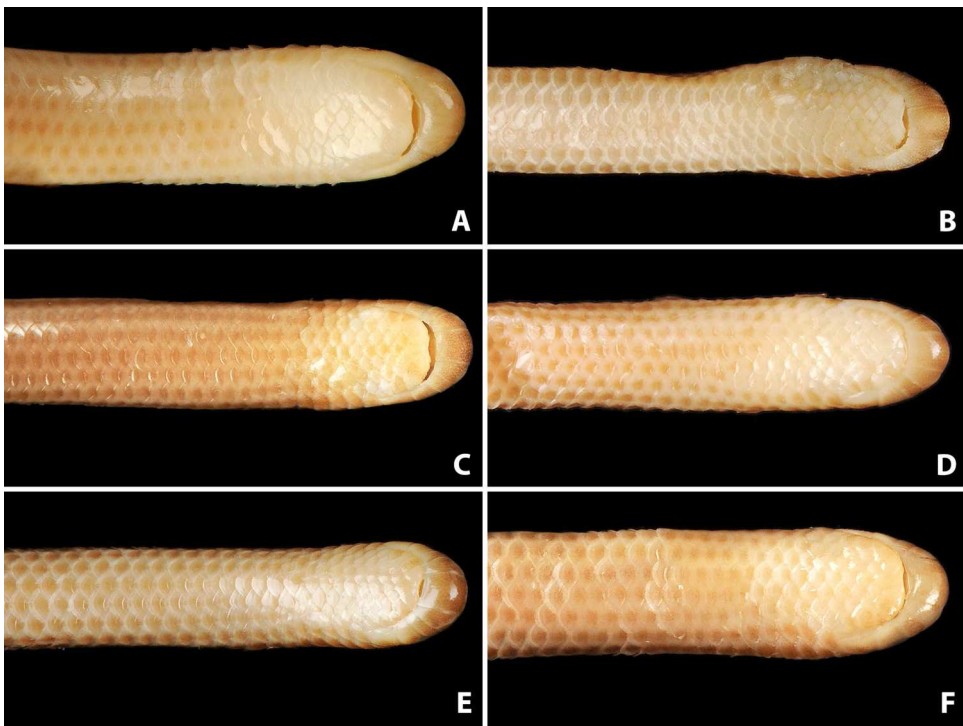

**Figure 5 Comparison of ventral views of the heads of holotype (NMW 15446:6, A) and paratypes of *Epictia rioignis* sp. nov. (B–F).** (A) Holotype (NMW 15446:6). (B) Paratype (NMW 15446:2). (C) Paratype (NMW 15446:4). (D) Paratype (NMW 15446:5). (E) Paratype (NMW 15446:7). (F) Paratype (NMW 15446:8). Photo credits: Alice Schumacher & Josef Muhsil.

rounded in dorsal, lateral and ventral views. Rostral subtrapezoidal in dorsal view with straight apex, straight in ventral view, reaching the imaginary transverse line between the anterior borders of the eyes, contacting upper and lower nasals laterally and frontal dorsally.

Nasal completely divided horizontally by an oblique suture, reaching rostral and first supralabial; ellipsoid nostril located in the center of the suture between upper and lower nasal, having the major axis obliquely oriented along the suture; supranasal higher than wide, contacting rostral anteriorly, infranasal inferiorly, first supralabial and supraocular posteriorly, and frontal dorsally; infranasal not visible in dorsal view, contacting rostral anteriorly and first supralabial posteriorly; two supralabial scales positioned anterior and posterior to ocular scale (1 + 1), respectively, resulting in an upper lip border formed by rostral, infranasal, anterior supralabial, ocular, and posterior supralabial; first supralabial 2.2 times higher than wide, exceeding nostril, reaching central level of eye, dorsally acuminate and in contact with supraocular scale; second supralabial subtrapezoidal 1.3 times higher than wide, slightly exceeding central level of eye, about as high as and at widest point 1.7 times wider than first supralabial; posterior margin of second supralabial in broad contact with temporal and first scale of lateral body row, dorsal margin in contact with parietal; temporal scale of same size as dorsal scales of lateral rows, but distinct from

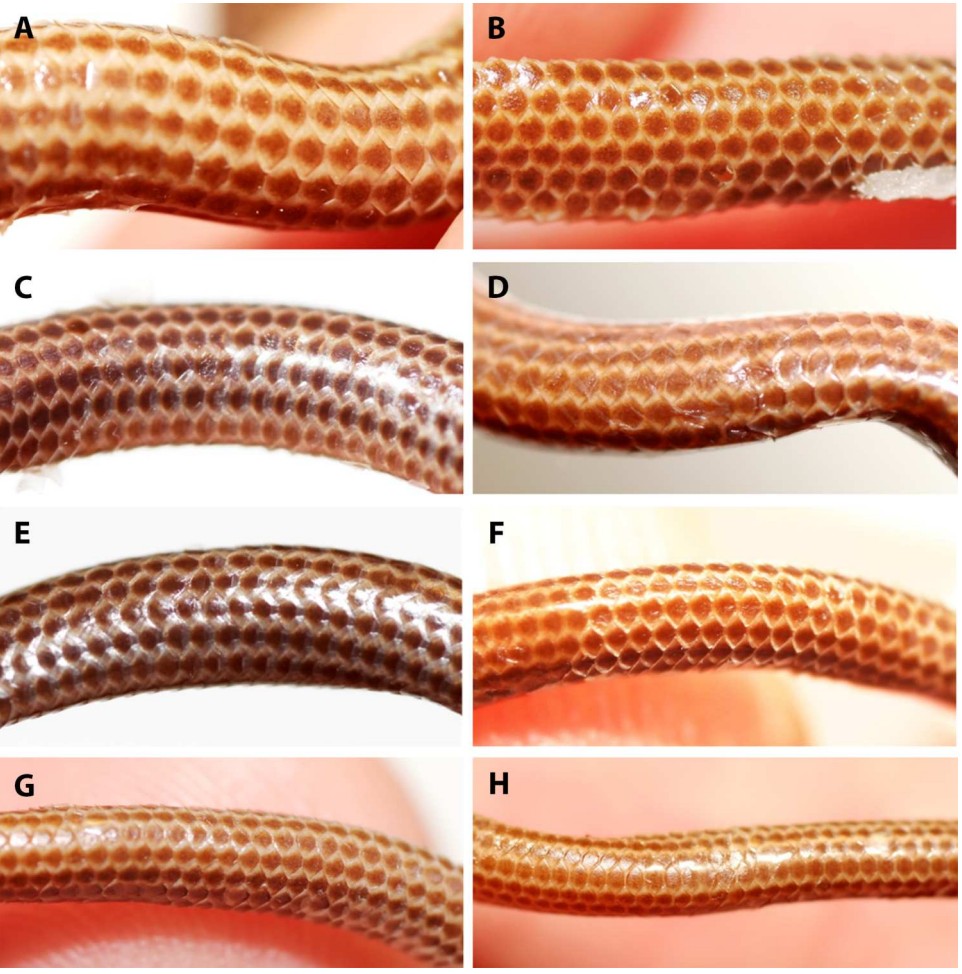

**Figure 6** **Comparison of dorsal body scales of holotype (NMW 15446:6, A) and paratypes of *Epictia rioignis* sp. nov. (B–H).** (A) Holotype (NMW 15446:6). (B) Paratype (NMW 15446:1). (C) Paratype (NMW 15446:2). (D) Paratype (NMW 15446:3). (E) Paratype (NMW 15446:4). (F) Paratype (NMW 15446:5). (G) Paratype (NMW 15446:7). (H) Paratype (NMW 15446:8).

lateral body scales by its oblique orientation; ocular scale pentagonal with dorsal apex acuminate and anterior border slightly rounded at eye-level, 1.8 times higher than wide, contacting anteriorly first supralabial, anterodorsally supraocular, posterodorsally parietal and dorsally second supralabial; eye distinct, located at level of maximum width of ocular, with lower eye margin at nostril level, positioned anteriorly and almost contacting scale sutures; upper half of eyes visible in dorsal view; supraocular scale oriented obliquely, about twice as long as wide, contacting supranasal anteriorly, parietal and postfrontal posteriorly, frontal dorsally, and first supralabial inferiorly; supraocular, parietal and occipital scales visible in lateral view; middorsal head plates (frontal, postfrontal, interparietal, and interoccipital) imbricate, subhexagonal, except for subtriangular frontal, with frontal and interoccipital being slightly larger than the other two scales; middorsal head plates narrower than posterior middorsal scales; frontal contacting rostral, supranasals, supraoculars, and

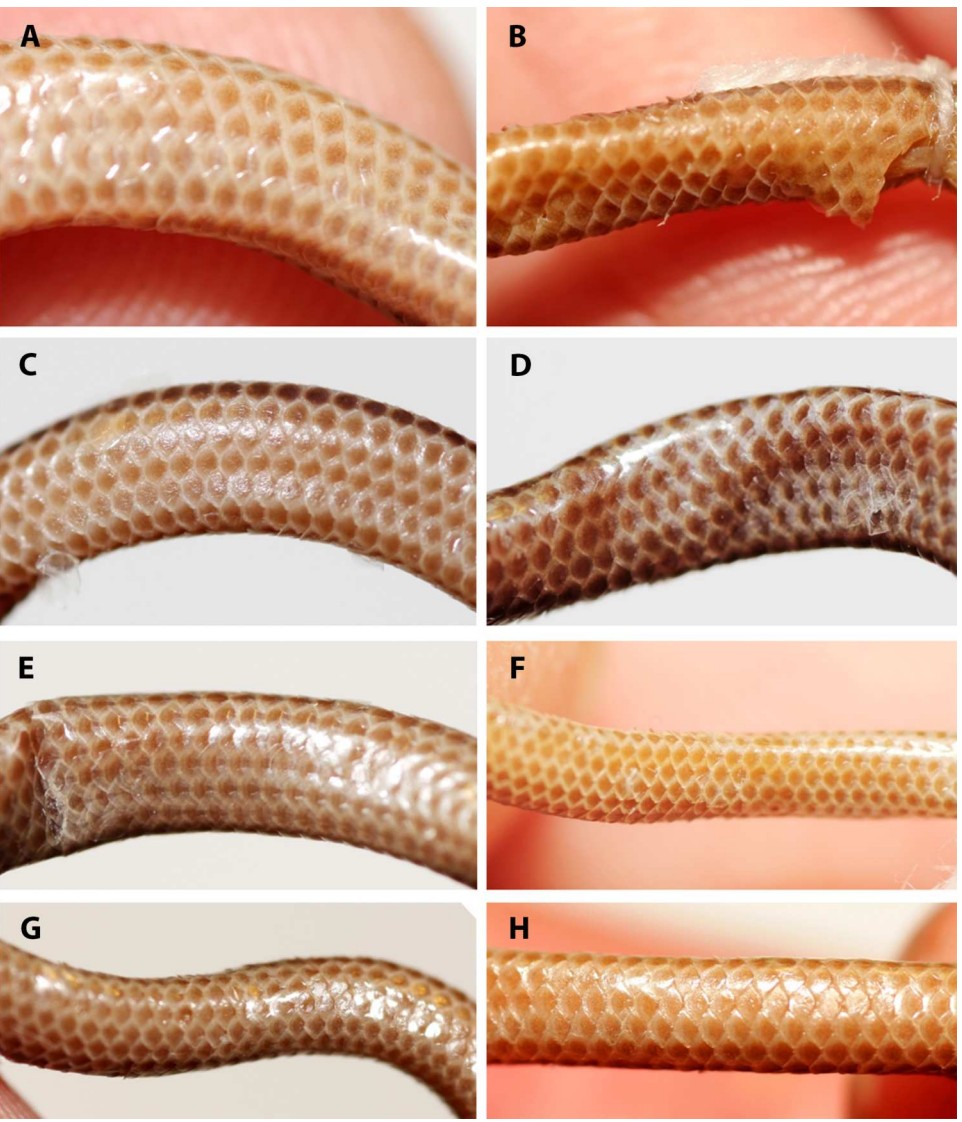

**Figure 7 Comparison of ventral body scales of holotype (NMW 15446:6, A) and paratypes of *Epictia rioignis* sp. nov. (B–H).** (A) Holotype (NMW 15446:6). (B) Paratype (NMW 15446:1). (C) Paratype (NMW 15446:2). (D) Paratype (NMW 15446:3). (E) Paratype (NMW 15446:4). (F) Paratype (NMW 15446:5). (G) Paratype (NMW 15446:7). (H) Paratype (NMW 15446:8). Photo credits: Alice Schumacher & Josef Muhsil.

postfrontal; postfrontal contacting frontal, supraoculars, parietals, and interparietal; interparietal contacting postfrontal, parietals, occipitals, and interoccipital; interoccipital contacting interparietal, occipitals, nuchal and first pair of paravertebral dorsal scales; parietal slightly larger than occipital, both irregularly hexagonal and about twice as high as wide; lower margin of parietal contacting upper border of posterior supralabial and temporal, posterior margin in broad contact with occipital, dorsal margin contacting postfrontal and interparietal, anterior margin in broad contact with ocular and supraocular; lower margin of occipital contacting temporal and first scale of lateral body row, posterior

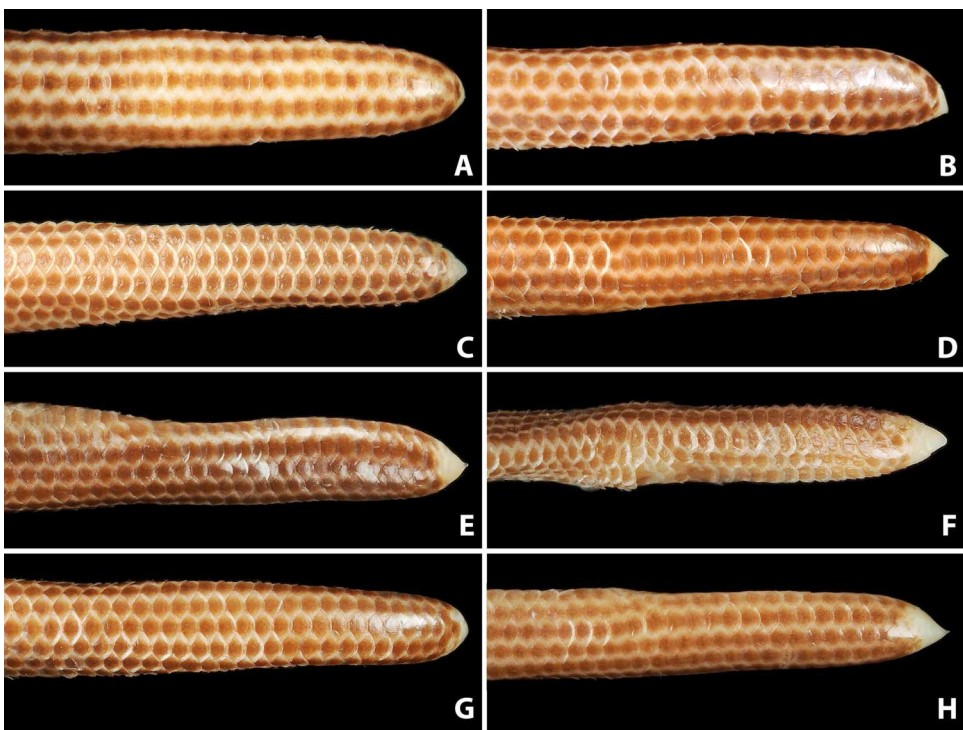

**Figure 8** Comparison of dorsal view of tails of holotype (NMW 15446:6, A) and paratypes of *Epictia rioignis* sp. nov. (B–H). (A) Holotype (NMW 15446:6). (B) Paratype (NMW 15446:1). (C) Paratype (NMW 15446:2). (D) Paratype (NMW 15446:3). (E) Paratype (NMW 15446:4). (F) Paratype (NMW 15446:5). (G) Paratype (NMW 15446:7). (H) Paratype (NMW 15446:8). Photo credits: Alice Schumacher & Josef Muhsil.

margin in broad contact with first paravertebral and first scale of dorsolateral body row, dorsal margin in contact with interparietal and interoccipital, anterior margin in broad contact with parietal; four infralabials per side, subequal in size, first three higher than wide, fourth wider than high, first two pairs of infralabials almost rectangular, larger than third infralabials; mental scale single, small, lunulate; labials, chin and gular scales, and dorsal and lateral head scales with numerous scattered pores.

Dorsal scales imbricate, smooth, homogeneous, rhomboid or elliptical in shape, about 1.5 times wider than long; 267 MDS; 14-14-14 D; 258 V; 10 TS. Cloacal shield large, subtriangular in shape, about 2.2 times wider than long, bordered anteriorly and posteriorly each by five scales; 15 SC, becoming successively narrower distally, no fused scales dorsally or ventrally on tail; terminal spine conical and shorter than wide.

**Color of holotype after more than 100 years of preservation in ethanol (Fig. 2, and upper left picture of Figs. 3–9)**

Dorsal head scales (supranasal, frontal, supraoculars, postfrontal, parietals, interparietal, occipitals, interoccipital) except for rostral, reddish-brown with cream-colored sutures; rostral reddish-brown in posterior part and with a small cream-colored blotch on the anterodorsal part, lower (ventral) part of rostral light greyish-brown; infranasal and

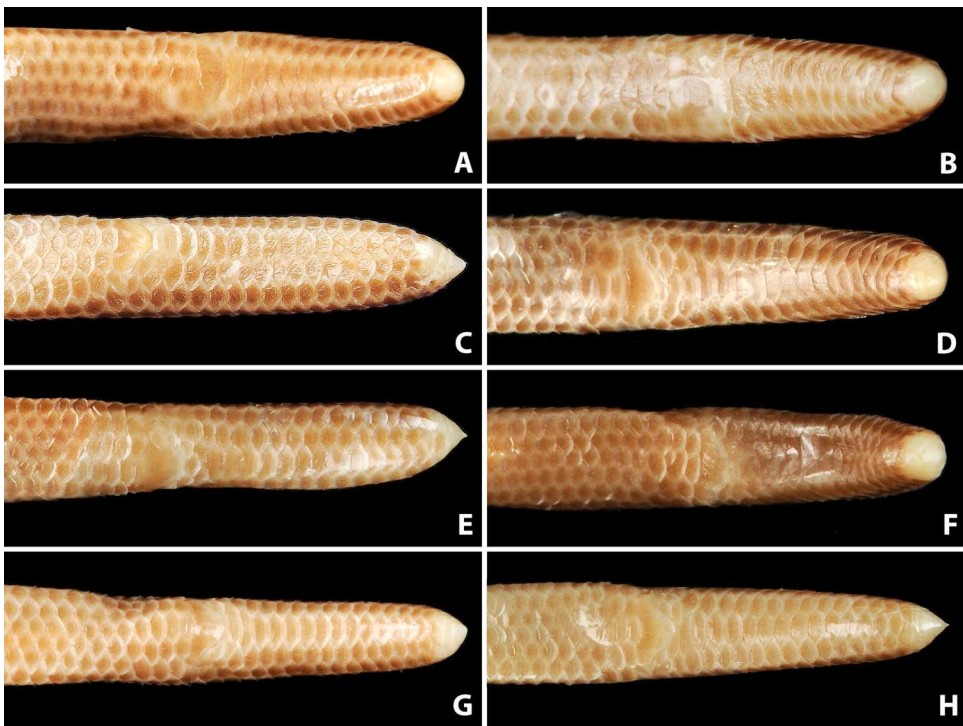

**Figure 9** Comparison of ventral view of tails of holotype (NMW 15446:6, A) and paratypes of *Epictia rioignis* sp. nov. (B–H). (A) Holotype (NMW 15446:6). (B) Paratype (NMW 15446:1). (C) Paratype (NMW 15446:2). (D) Paratype (NMW 15446:3). (E) Paratype (NMW 15446:4). (F) Paratype (NMW 15446:5). (G) Paratype (NMW 15446:7). (H) Paratype (NMW 15446:8).

anterior supralabial light reddish-brown; ocular scale mostly cream-colored; posterior supralabial light reddish-brown in upper half and cream-colored in lower half; ventral head scales (mental, infralabials, scales of chin and gular region) cream-colored; each dorsal body scale with a more or less triangular, median to dark reddish-brown blotch in the center of the scale and with cream-colored lateral margins, forming a pattern of dark longitudinal stripes with cream-colored interspaces, dark stripes slightly broader than interspaces, reaching to the penultimate scale of the tail; each ventral body scale, anal plate and scales on ventral part of tail light or median brown in central part and with cream-colored margins; terminal spine cream-colored.

## Osteology of holotype
### Skull (Figs. 10 and 11)
Premaxilla roughly rectangular in anterior view and hexagonal in ventral view, edentulous, pierced by six foramina (two in anterior view and four in ventral view); transverse process of premaxilla absent and vomerian process single; premaxilla with internal septum composed by two laminae that support the *septum nasii* dorsally, expanding posteriorly to fit medially in the septomaxilla (internally); nasals paired, approximately rectangular in dorsal view, being pierced by a pair of foramina in lateral border of contact with prefrontals (foramen for the *apicalis nasi*); an additional pair of foramina pierce the medial contact within

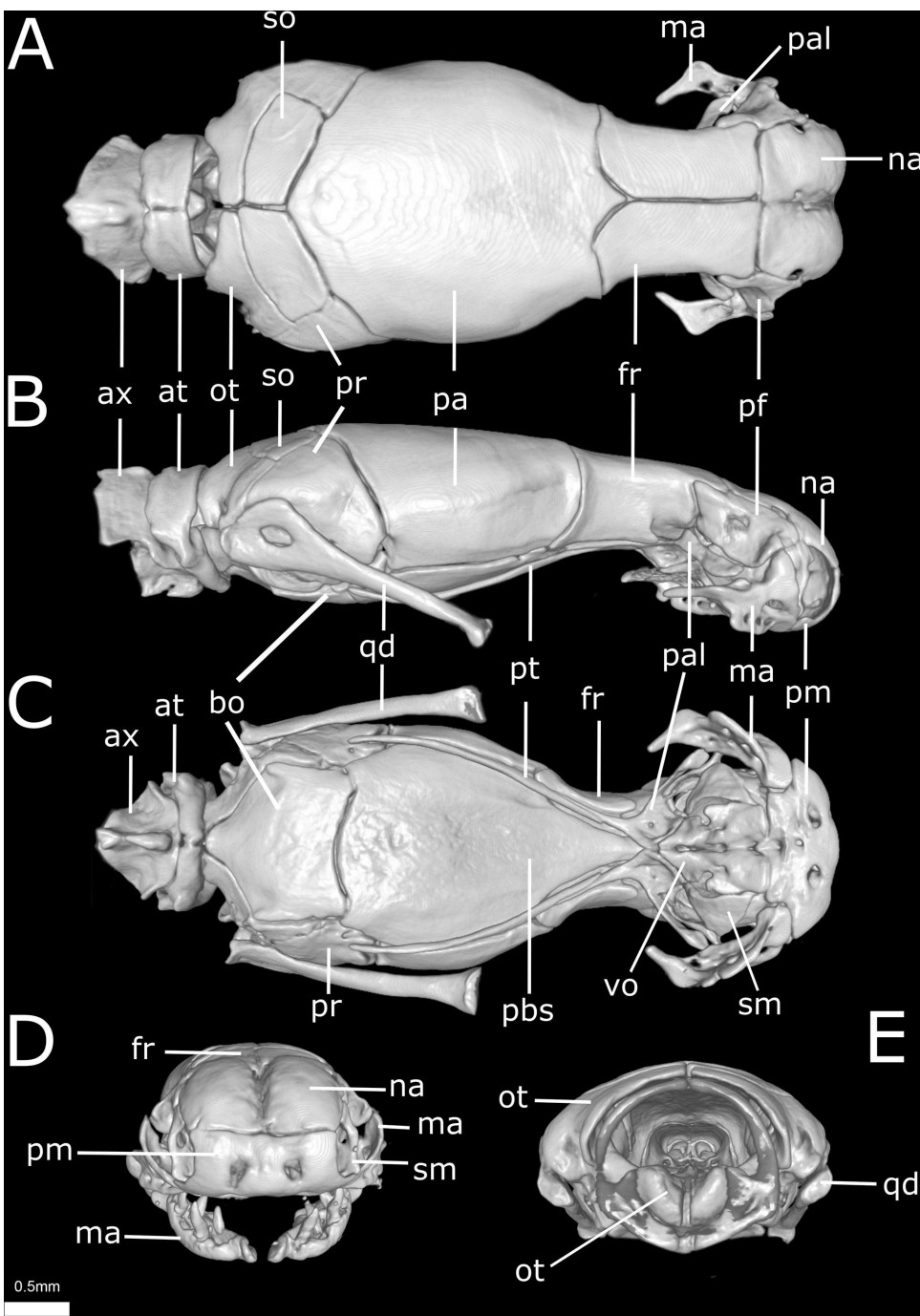

**Figure 10  Three-dimensional reconstruction of the skull of *Epictia rioignis* sp. nov. based on Micro-CT data of the holotype (NMW 15446:6).** (A) Dorsal, (B) lateral, (C) ventral, (D) anterior, and (E) posterior views. at, atlas; ax, axis; bo, basioccipital; fr, frontal; ma, maxilla; na, nasal; ot, otooccipital; pa, parietal; pal, palatine; pbs, parabasisphenoid; pf, prefrontal; pm, premaxilla; pr, prootic; pt, pterygoid; qd, quadrate; sm, septomaxilla; so, supraoccipital; vo, vomer.

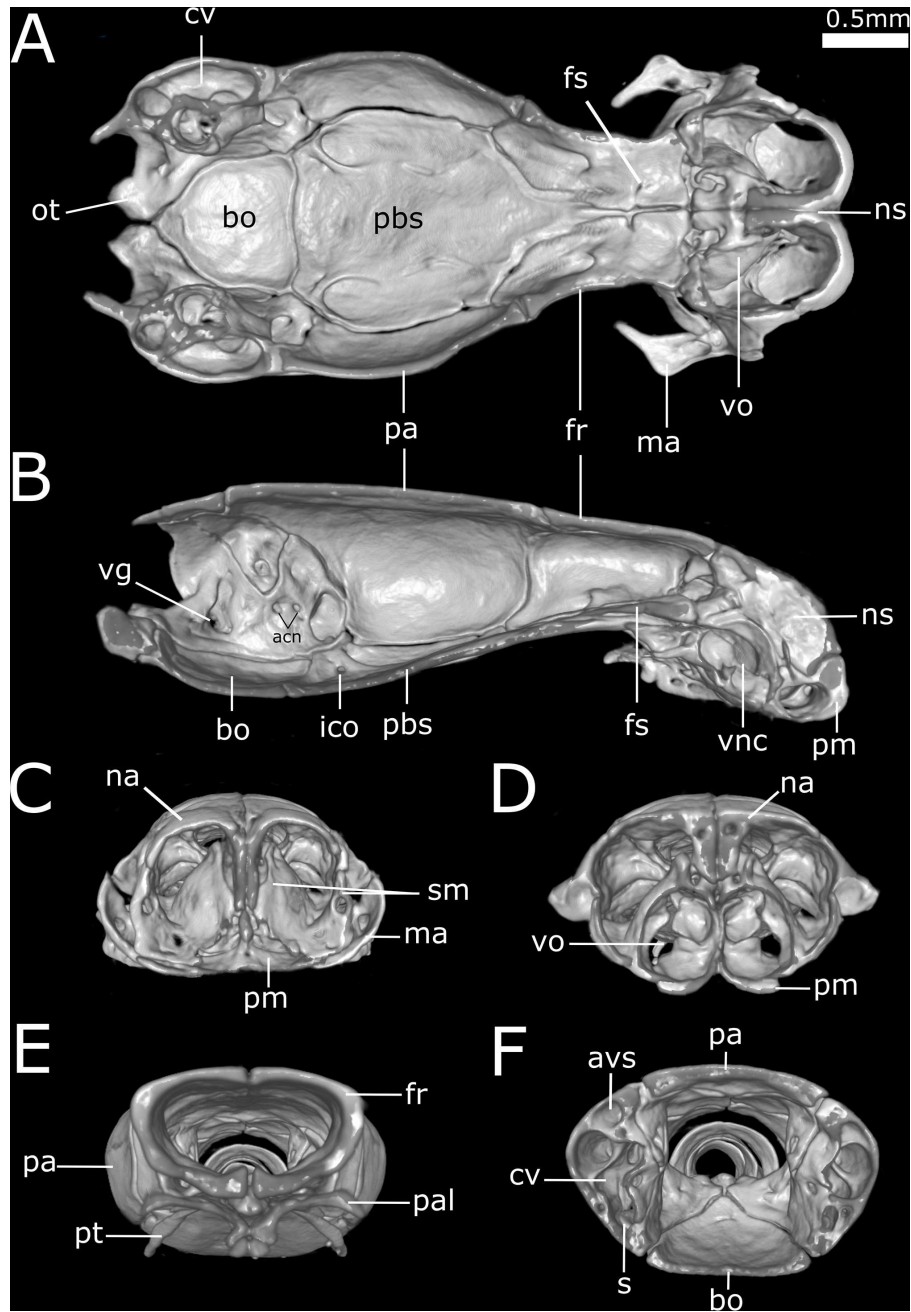

**Figure 11 Three-dimensional cutaway views of the skull of *Epictia rioignis* sp. nov. based on Micro-CT data of the holotype (NMW 15446:6).** Cuts along the (A) frontal, (B) sagittal and (C–F) transverse axis. acn, acoustic nerve; avs, anterior vertical semicircular canal; bo, basioccipital; cv, *cavum vestibuli*; fs, frontal subolfactory process; fr, frontal; ico, internal carotid opening; ma, maxilla; na, nasal; ns, nasal septum; ot, otooccipital; pa, parietal; pal, palatine; pbs, parabasisphenoid; pm, premaxilla; pt, pterygoid; s, stapes; sm, septomaxilla; vg, vagus nerve foramen; vnc, vomeronasal cupola; vo, vomer.

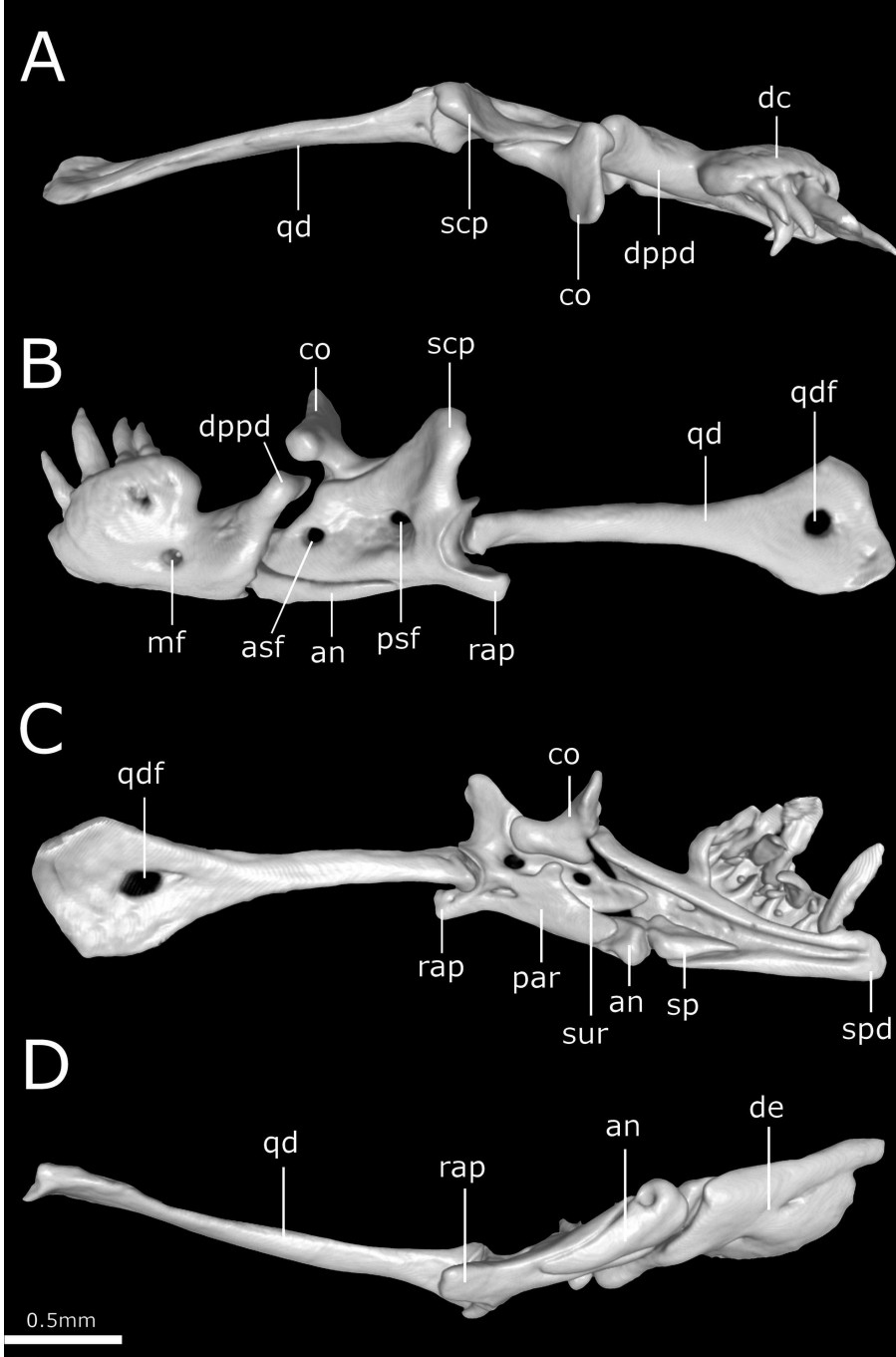

**Figure 12  Three-dimensional reconstruction of the suspensorium (quadrate + lower jaw) of *Epictia rioignis* sp. nov. based on Micro-CT data of the holotype (NMW 15446:6).**  (A) Dorsal, (B) lateral, (C) medial, and (D) ventral views. an, angular; asf, anterior surangular foramen; co, coronoid; dc, dental concha; de, dentary; dppd, dorsoposterior process of dentary; mf, mental foramen; par, prearticular lamina of compound bone; psf, posterior surangular foramen; qd, quadrate; qdf, quadrate foramen; rap, retroarticular process; scp, supracotylar process of surangular; sp, splenial; spd, symphyseal process of dentary; sur, surangular lamina of compound bone.

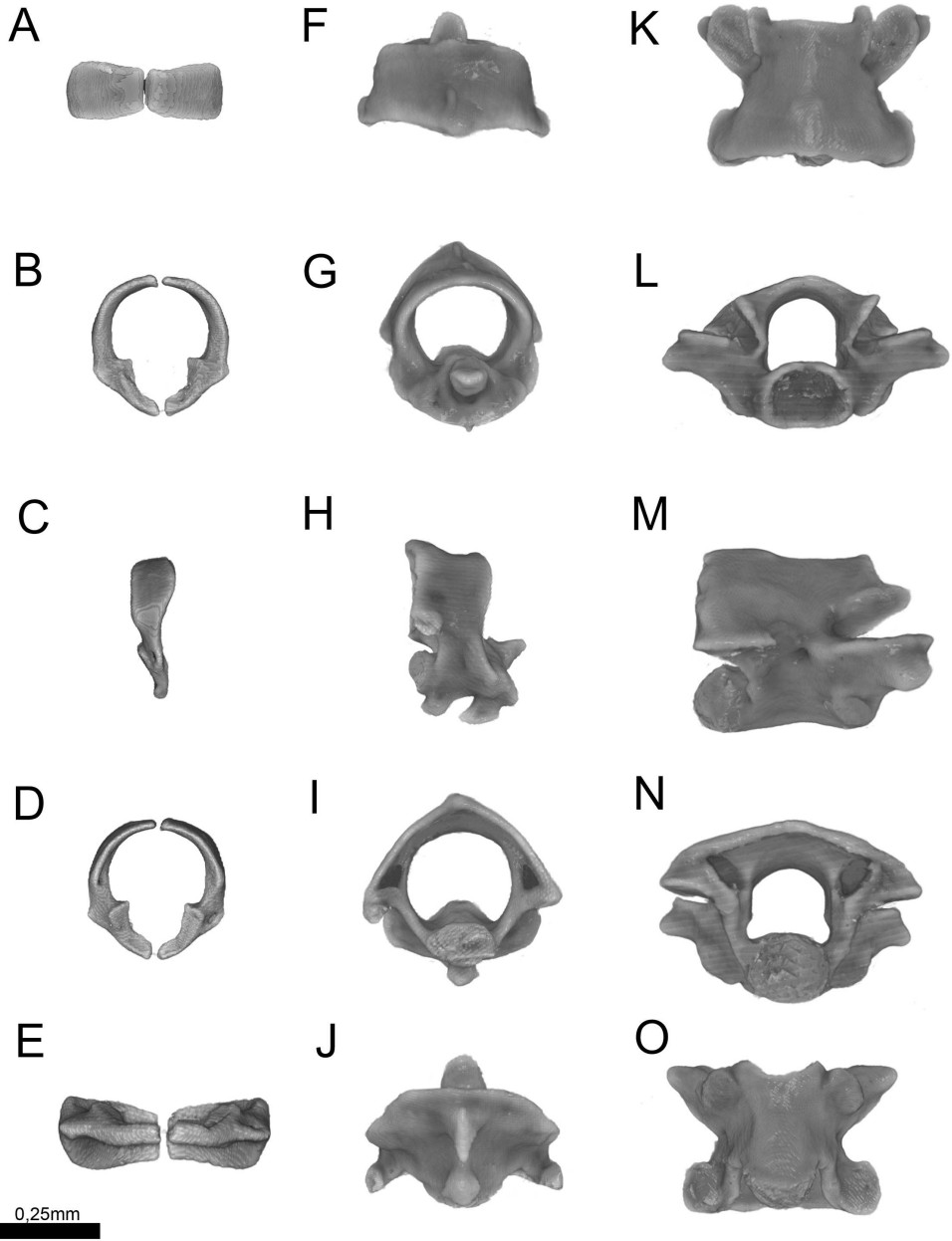

0,25mm

**Figure 13** **Three-dimensional reconstruction of the (A–E) atlas, (F–J) axis and (K–O) midtrunk vertebrae of *Epictia rioignis* sp. nov. based on Micro-CT data of the holotype (NMW 15446:6).** (A, F, K) Dorsal, (B, G, L) anterior, (C, H, M) lateral, (D, I, N) posterior, and (E, J, O) ventral views.

both nasals; a single additional foramen pierces the anterior-medial region of the nasal dorsal lamina; nasal septum descending as double medial vertical flanges that contact the premaxilla, septomaxilla and vomer ventrally (internally); prefrontals paired, subtriangular in dorsal view, in contact with septomaxilla and maxilla ventrally; septomaxillae paired, complex in shape, expanding dorsally into the naris; conchal invagination absent; ascending process of premaxilla pierced by single large foramen; internally, dorsal surface of each

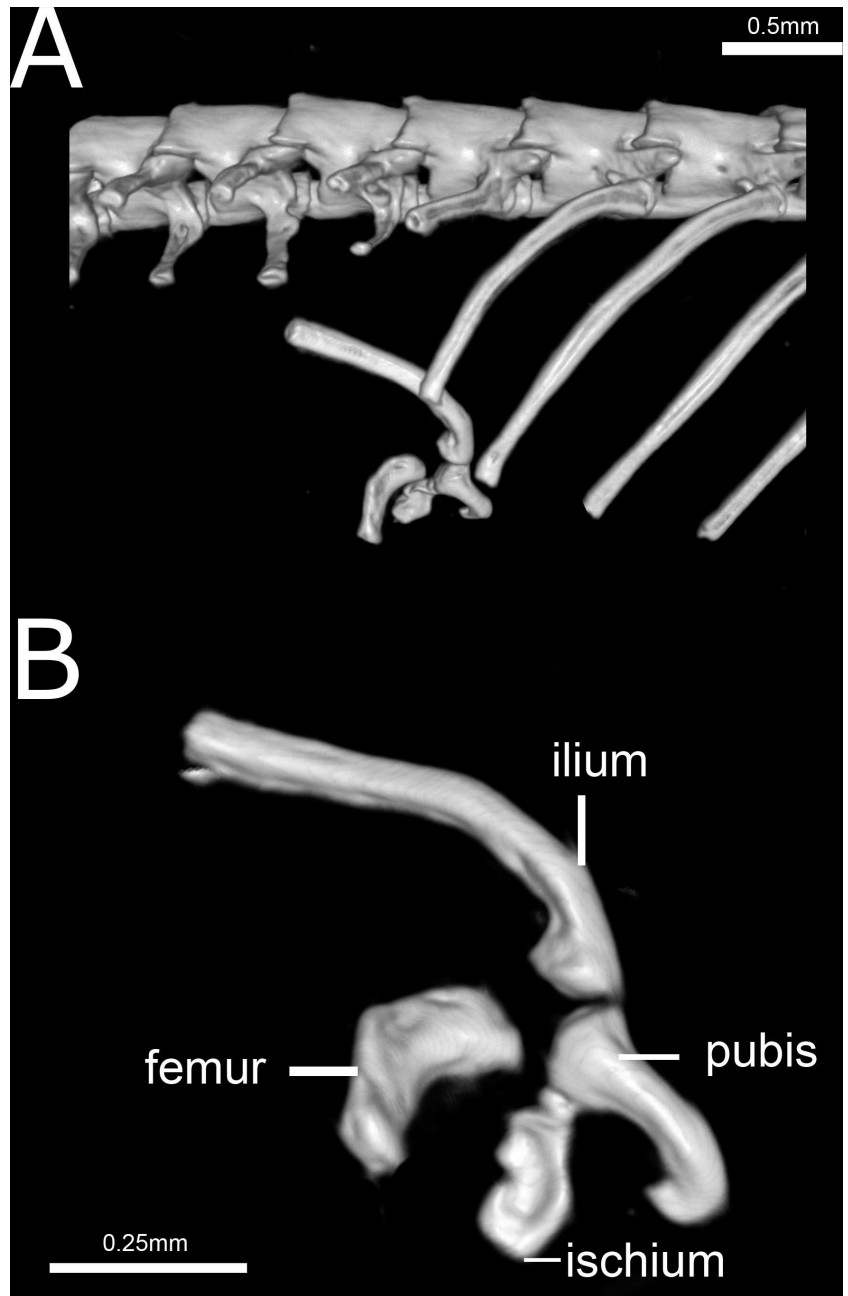

**Figure 14** **Three-dimensional reconstruction of the pelvic girdle of *Epictia rioignis* sp. nov. based on Micro-CT data of the holotype (NMW 15446:6).** (A) Overview with position and orientation. (B) Pelvic girdle digitally isolated in lateral view.

septomaxilla pierced by a foramen, and with a medial deep sulcus that extends from its posterior to anterior regions; vomers paired, located midventral to vomeronasal cupola, bearing transversal arms, and with short posterior arms in contact with each other posteriorly; a pair of foramina pierce the ventral lamina of the vomer; frontals paired, nearly rectangular in dorsal view, the left element bearing short anterolateral projections

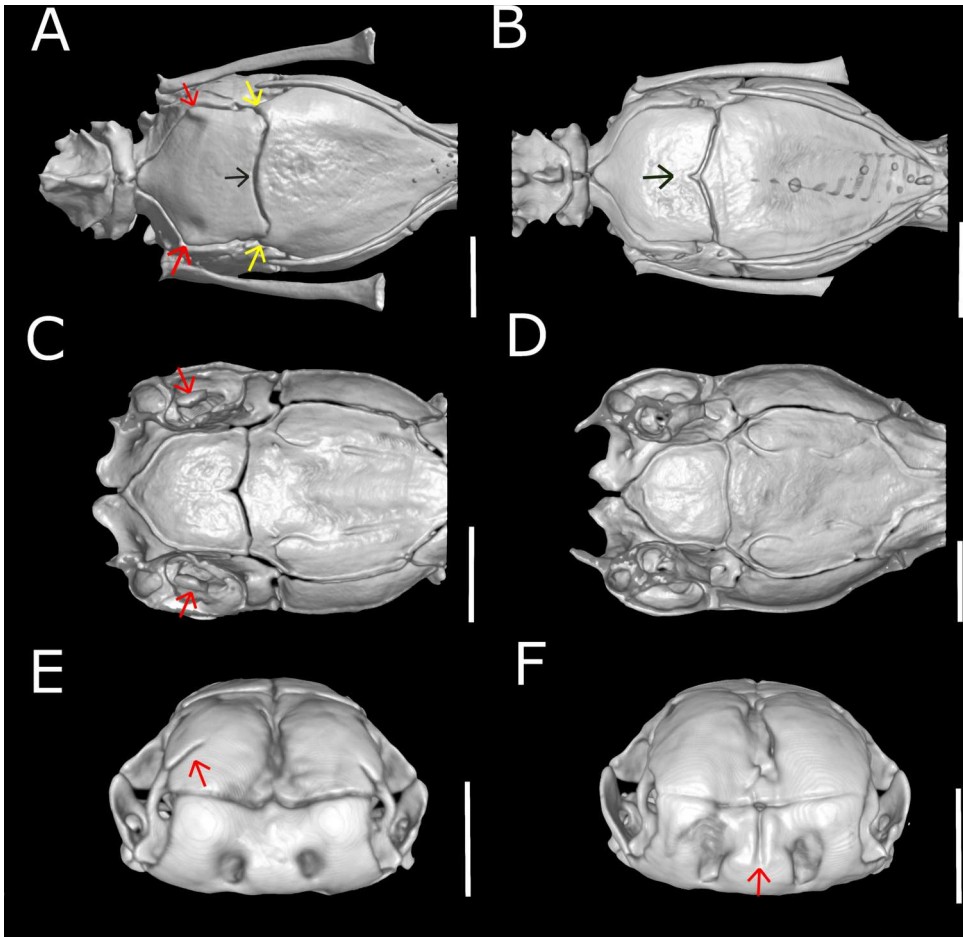

**Figure 15** **Three-dimensional reconstruction of skulls of different specimens of *Epictia rioignis* sp. nov. based on Micro-CT data, showing variations in skull parameters.** (A) Ventral views showing shape variation of basioccipital of holotype (NMW 15446:6) lacking medial recess and (B) paratype (NMW 15446:7) with medial recess. (C) Cutaway views along the frontal axis of paratype (NMW 15446:7) with presence of statolythic mass and (D) absence in the holotype (NMW 15446:6). (E) Anterior views of paratype (NMW 15446:4) with lateral recess of nasal and (F) paratype (NMW 15446:8) with medial sulcus of premaxilla.

to attach to prefrontals; frontal pillars absent; optic nerve restricted to lateral descending surface of frontals; maxilla edentulous, irregular in shape, pierced by four large foramina in lateral view, two in the dentigerous process of maxilla; posterior process of maxilla reaching the level of the optic nerve foramen; posterior orbital element absent; parietal single, wide, representing the largest bone of braincase; parietal internal pillars (*sensu Martins, 2016*) absent; parabasisphenoid arrow-like, with tapered anterior tip lying dorsally to palatine, and fitting in medial line of vomeronasal cupola; in ventral view, parabasisphenoid bearing posterior-lateral projections to provide insertion for the neck muscles (*Martins, Passos & Pinto, 2019*); parabasisphenoid with shallow pituitary fossa and lateral sulcus; anterior opening for the palatine artery indistinct or absent in parabasisphenoid dorsal (internal) surface, internal carotid artery foramen and *abduscens* nerve foramen present; opening for

**Table 1 Morphometrics, pholidosis and osteological characters of *Epictia rioignis* sp. nov.**

| No. | MDS | V | PCV | SC | CAV | CLV | TL (mm) | TAL (mm) | MB (mm) | MT (mm) | HL (mm) | HW (mm) | HH (mm) | ED (mm) |
|---|---|---|---|---|---|---|---|---|---|---|---|---|---|---|
| NMW 15446:1 | / | / | / | 14 | / | / | / | 9.3 | 3.4 | 2.6 | / | / | / | / |
| NMW 15446:2 | 250 | 237 | 231 | 19 | 20 | 4 | 113 | 7.7 | 2.6 | 1.8 | 2.6 | 2 | 1.3 | 0.4 |
| NMW 15446:3 | / | / | / | 19 | / | / | / | 9.2 | 3.4 | 2.8 | / | / | / | / |
| NMW 15446:4 | 261 | 243 | 235 | 17 | 18 | 4 | 149 | 8.2 | 3 | 2.4 | 2.9 | 2.2 | 1.9 | 0.5 |
| NMW 15446:5 | 262 | 245 | 246 | 18 | / | / | 110 | 5.9 | 2 | 1,7 | 2.4 | 2 | 1.4 | 0.4 |
| NMW 15446:6 | 267 | 258 | 248 | 15 | 17 | 4 | 211 | 10.3 | 4.4 | 3.1 | 3.6 | 2.8 | 2.3 | 0.4 |
| NMW 15446:7 | 265 | 248 | 240 | 18 | 14 | 5 | 157 | 7.9 | 2.7 | 2.2 | 2.7 | 2.2 | 1.6 | 0.4 |
| NMW 15446:8 | 260 | 252 | 238 | 18 | 21 | 3 | 157 | 9.3 | 2.8 | 2.1 | 2.9 | 2.4 | 1.8 | 0.4 |

**Notes.**

CAV, caudal vertebrae; CLV, cloacal vertebrae; ED, eye diameter; HH, head height; HL, head length; HW, head width; MB, midbody diameter; MDS, middorsal scale rows; MT, midtail diameter; PCV, precloacal vertebrae; SC, subcaudals; TAL, tail length; TL, total length; V, ventral scales.

the palatine ramus of the facial nerve formed by the lateral edge of the parabasisphenoid and the ventral edge of the parietal; basioccipital single and approximately pentagonal in ventral view, bearing lateral process to attach tendons for the neck muscles (A Martins, 2016, unpublished data); basioccipital does not participate in the formation of the foramen magnum; supraoccipitals paired, approximately rectangular in dorsal view, pierced medially (internally) by a large endolymphatic foramen; prootics paired and triangular in lateral view; prootics forming the trigeminal nerve foramen together with the parietal; prootics pierced medially (internally) by two acoustic nerve foramina, and an additional foramen ventral to the former; statolythic mass in *cavum vestibuli* absent; stapedial footplate apparently not co-ossified with prootic; otooccipitals paired and irregular in dorsal view, descending to contact each other ventrally to exclude the basioccipital in the formation of the foramen magnum and forming a short but distinct atlantal process (*sensu Cundall & Irish, 2008*); medial surface (internal) of otooccipitals pierced by an internal opening for the *recessus scalae timpani* and a wide foramen that forms the internal opening for the *vagus* nerve foramen; a reduced foramen pierces the posterior (external) surface of the otooccipital, posterior to the external opening for the *vagus* nerve foramen; palatines paired and triradiate; anterior margin of maxillary process flexing ventrally; palatines pierced by a foramen in its ventral surface; pterygoids slender and rod-like, not contacting quadrate posteriorly, and not extending beyond the anterior margin of basioccipital; ectopterygoid indistinct.

### *Suspensorium (Fig. 12)*

Dentary supports a series of six teeth ankylosed to the inner surface of the anterolateral margin of dental concha; mental foramen nearly under the 6th tooth; splenial conical, visible in lateral view, representing smallest bone in lower jaw, extending from the level of the 5th tooth to contact the angular posterior; anterior mylohyoid foramen absent on splenial; posterior mylohyoid foramen on the ventral and dorsal surface of angular; angular conical, extending posteriorly to level of the posterior surangular foramen; compound bone pierced by two foramina in the surangular lamina which are approximately similar in size,

anterior surangular foramen located anterior to the coronoid; foramen for the chorda tympani of the hyomandibular ramus of the facial nerve (VII) present; a small foramen pierces the retroarticular process in medial view; prearticular lamina of compound bone presenting dorsal process to support the coronoid; quadrate long and slender, about 50% of skull length, presenting a posterior process (*sensu Martins, 2016*); a dorsal foramen in the anterior half of quadrate absent.

### Cervical vertebrae (Fig. 13)

Atlas composed by neural arches, not fused dorsally but fused ventrally to a reduced ventral element (*intercentrum I* sensu *Holman, 2000*); short lateral projections of the atlas are present; axis with short spinal process; lateral foramina of axis indistinct or absent; short lateral processes present. Odontoid process of axis osseous and sutured to axis, approximately losangular in anterior view, with an anterior tapered process; intercentra II and III ventral, compressed laterally, bearing hypapophysis pointed in lateral view (sensu *Holman, 2000*).

### Trunk vertebrae (Fig. 13)

Trunk vertebrae at midbody dorsoventrally flattened and bearing ribs; neural arches flattened; neural spines absent; spinal processes short and distinct; neural canal in anterior view about as high as wide; epyzygapophyseal spines absent; zygosphenes present, emerging from the neural canal; zygosphenal crests laterally expanded and tapered distally; zygosphenal articular facets absent; prezygosphenal articular facets ellipsoidal; prezygapophyses expand into prezygapophyseal accessory processes, distally rounded and visible in dorsal view; zygantrum wide laterally, limited by the relatively developed zygantral articular facets; postzygapophyses laterally expanded and flattened; cotyle ellipsoidal in anterior view; supracotilar and paracotylar foramina indistinct or absent; condyle ellipsoidal; vertebrae centrum bearing lateral foramen at each side; synapophyses with no distinction between diapophyseal and parapophyseal areas; hemal keel absent; subcentral ridges absent.

### Pelvic girdle (Fig. 14)

Composed by ilium, ischium, femur, and pubis. Ilium and pubis rod-like; ischium approximately rectangular, apparently fused to pubis; ilium represents the longest bone of pelvic girdle; femur approximately rectangular and curved, slightly larger than ischium, located at between the last trunk vertebrae and second cloacal vertebrae.

## Variation

The paratype series consists of seven specimens, two of which are lacking the head and anterior part of body (NMW 15446:1 and NMW 15446:3). Thus information of measurements and scalation of the tail and the coloration of body and tail is based on all specimens, whereas information on other characters (e.g head measurements, skull conditions) is based only on the complete specimens.

### Variation of scale counts and measurements (see Table 1 for individual data of each specimen)

250–267 MDS ($\bar{x} = 260.83 \pm 5.91$, $n = 6$); 237–258 V ($\bar{x} = 247.17 \pm 7.31$, $n = 6$); 14–19 SC ($\bar{x} = 17.25 \pm 1.83$, $n = 8$); TL of 110–211 mm ($\bar{x} = 149.5 \pm 36.84$, $n = 6$); TAL of 5.9–10.3 mm ($\bar{x} = 8.48 \pm 1.36$, $n = 6$); MB of 2–4.4 mm ($\bar{x} = 3.04 \pm 0.71$, $n = 8$); MT of 1.7–3.1 mm ($\bar{x} = 2.34 \pm 0.48$, $n = 6$); TL/TAL of 14.7–20.5 ($\bar{x} = 18.13 \pm 2.11$, $n = 6$); TL/MB of 43.5–58.1 ($\bar{x} = 51.73 \pm 5.59$, $n = 6$); TAL/MT of 3.3–4.3 ($\bar{x} = 3.66 \pm 0.41$, $n = 8$); HW of 2–2.8 mm ($\bar{x} = 2.27 \pm 0.30$, $n = 6$); HH of 0.4–0.5 mm ($\bar{x} = 0.417 \pm 0.04$, $n = 6$); HL of 2.4–3.6 mm ($\bar{x} = 2.85 \pm 0.41$, $n = 6$); ED of 1.3–2.3 mm ($\bar{x} = 1.717 \pm 0.37$, $n = 6$); RES of 0.3–0.4 ($\bar{x} = 0.365 \pm 0.35$, $n = 6$).

### Variation of color pattern

Color pattern of paratypes mostly resembles that of the holotype except for the following (Figs. 3–9): no cream-colored blotch on anterodorsal part of rostral recognizable in one paratype (NMW 15446:4), cream-colored blotch on rostral indistinct in two paratypes (NMW 15446:7, NMW 15446:8), cream-colored blotch more distinct and larger than in holotype, covering more than half of rostral in dorsal view in one paratype (NMW 15446:5); ocular scale almost entirely light to median reddish-brown in all paratypes; posterior supralabial almost entirely reddish brown in one paratype (NMW 15446:4); some infralabials and some adjacent chin scales exhibit a light brown pigmentation in some paratypes; in some paratypes the cream-colored ultimate part of the tail covers also the scales adjacent to the terminal spine.

### Qualitative and quantitative variation of skull and lower jaw ($n = 6$; holotype condition indicated with asterisk)

Dentary teeth 5 ($n = 3$; 50%) or 6* ($n = 3$; 50%); premaxilla pierced by two foramina in anterior view and two large foramina in ventral view ($n = 1$), three in ventral view and two in anterior view ($n = 1$) or two in anterior view and four in ventral view ($n = 5$*); medial sulcus of premaxilla present ($n = 1$) or absent* ($n = 5$); maxilla perforated by one ($n = 1$), three ($n = 1$), four* ($n = 2$), five ($n = 1$) or six ($n = 1$) foramina; projections for the attachment of neck muscles in parabasisphenoid absent ($n = 5$; Fig. 15B) or present* ($n = 1$; Fig. 15A, yellow arrows), projections for the attachment of neck muscles in basioccipital absent ($n = 5$; Fig. 15B) or present* ($n = 1$; Fig. 15A, red arrows); statolythic mass present ($n = 3$; Fig. 15C) or absent* ($n = 3$; Fig. 15D); basioccipital with medial recess ($n = 2$; Fig. 15B) or not* ($n = 4$; Fig. 15A, black arrow); lateral recess of nasal present ($n = 1$; Fig. 15E) or absent* ($n = 5$; Fig. 15F).

### Postcranial quantitative variation

Precloacal vertebrae 231–248 ($\bar{x} = 240 \pm 6.5$, $n = 6$); caudal vertebrae 14–21 ($\bar{x} = 18 \pm 2.7$, $n = 5$). Correlation ($n = 5$) between middorsal scales and precloacal + cloacal + caudal vertebrae (0.99:1), between midventral scales and precloacal vertebrae (0.97:1), and between subcaudal scales and cloacal + caudal vertebrae (1:1.3). Pelvic girdle located at the level of the 229th (=penultimate) trunk vertebrae and first cloacal vertebrae (NMW 15446:2), 233–234th (=penultimate and last) trunk vertebrae (NMW 15446:4), 246th

(=last) precloacal and second cloacal vertebrae (NMW 15446:5), 248th (= last) trunk and second cloacal vertebrae (NMW 15446:6), 240th (=last) trunk and first cloacal vertebrae (NMW 15446:7) and 238th (=last) trunk and second cloacal vertebrae (NMW 15446:8).

## Postcranial qualitative variation ($n = 6$; holotype condition indicated with asterisk)

Ventral element of atlas absent or not ossified to neural arches ($n = 3$) or absent with neural arches fused ventrally* ($n = 3$); anterior hypapophysis of axis pointed* ($n = 5$) or rounded ($n = 1$); posterior hypapophysis of axis rounded ($n = 1$), truncated ($n = 3$) or pointed* ($n = 2$); hypapophysis of axis fused ($n = 1$) or not* ($n = 5$); femur fused ($n = 1$) or not* ($n = 5$) to ischium.

## Etymology

The specific epithet is an agglutination of the Latin nomen "ignis" which means fire and the proper noun "Rio" as an acronym for the Brazilian city of Rio de Janeiro. This name was chosen in honour to the Museu Nacional do Rio de Janeiro/UFRJ, Brazil's oldest scientific institution with the largest South American collections of zoology, anthropology, geology and paleontology. Many of the precious collections pertaining to the zoology department (mostly invertebrates), anthropology, geology and paleontology were completely destroyed in the disastrous fire in its main building on September 2nd 2018. Due to historical neglection of this institution from the Brazilian government, added with substantial funding decrease in the past 5 years the museum did not receive sufficient money to fullfil basic safety standards—such as fire protection. The description of this new species, with specimens housed in a scientific collection for more than 100 years highlights one of the several importances of zoological collections in housing relevant material to understand the diversity of life, and also reinforce that such collections are timeless treasures for science. Such collections should receive strong attention in government investments as they contribute to the global development of science.

## Distribution and natural history

*Epictia rioignis* is currently known exclusively from its type series, from Corinto, Nicaragua (See comments under Discussion).

## DISCUSSION

Steindachner (1834–1919) was curator of Ichthyology and Herpetology in the NMW from 1860–1919. On the original label of NMW 15446 (Fig. 16A) he only noted "Corinto, 1907, Steind. don". but failed to mention a country name. Unfortunately and in addition, Steindachner was not always consistent with respect to the event, to which the year mentioned on his labels refered to. In some occasions the year refers to the collecting event, whereas in other occasions the year refers to the acquisition of the specimen(s) or even to the year the vouchers were inventoried by Steindachner (*Neumann, 2011*). At least, the color of the labels of the Herpetological collection always referred to a certain continent. The green labels were used for "America" and thus one can conclude that this is the continent of origin of the voucher specimens NMW 15446. Later Eiselt (curator of the

herpetological collection from 1952–1977) added "S-Amerika" on a newer label (Fig. 16B) and likewise mentioned "Corinto, S-Amerika, 1907" in the inventory book. Considering the trips of Steindachner and the specimens in the ichthyological and herpetological collection of the NMW, the name Corinto may refer to at least three localities in South- or Central America: Corinto, Minas Gerais (Brazil), Corinto, Cauca (Colombia), and Corinto (Nicaragua) (http://geonames.nga.mil/namesgaz). Corinto, Minas Gerais (Brazil) was named "Curralinho" at the time when the specimens were collected. In 1923, and thus many years after the collecting event, it became a municipality and was renamed as Corinto (*Barbosa, 1995*). Thus, if Steindachner had acquired the specimens from this locality he would have assigned them to "Curralinho" rather than Corinto. Corinto, Cauca (Colombia) is a town and municipality in the Cauca department (29,308 km$^2$). In the collections of the NMW are indeed specimens from the "Cauca region" inventoried by Steindachner, but the city of Corinto was never mentioned. An extended query for the name "Corinto" in the databases of the NMW yielded a single result: eight marine fish species from "Corinto, Nicaragua, collected in 1901, Schiff Donau" were found in the inventory of the fish collection. When Steindachner acquired several specimens from the same collecting event he often provided just one main label with further information (e.g., the country of origin) and used labels with reduced information for the other specimens, independent of the fact that the specimens belonged to different taxonomic groups (e.g., fishes and reptiles). As no other label for "Corinto" with more specific information could be found, we assume that the only detailed label found in the fish collection represents the "main" label of Steindachner for all specimens from the locality Corinto. Thus we further assume that the specimens of *Epictia rioignis* sp. nov. found in the Herpetological Collection of the NMW derive from the same collecting event and thus originate from Corinto in Nicaragua. The Nicaraguan Corinto is an important port town in the Northwest Pacific Coast which was founded in 1858, and has been historically important as a point of entry to Nicaragua since early 19th century. When labelling Corinto, Steindachner might have referred to both, the locality where the specimens were collected or the locality from where they were acquired. Therefore, considering the data provided herein, the locality of Corinto might still be considered with certain care, as the specimens might have been brought from other localities in Nicaragua.

Nicaragua is an important megadiverse country, however, its snake fauna is less diverse in comparison to other neighboring countries in Central America (*Sunyer, 2014*). Even if snake diversity is relatively low, studies on the Nicaraguan herpetofauna are still incipient and additional efforts on herpetological research might reveal novel data on the snake fauna of this country (*Sunyer, 2014*). So far, amongst *Epictia* spp., only *E. ater* has been previously reported to occur in Nicaragua, inhabiting lowland arid forests, premontane wet forests and dry tropical forests at elevations between 40–100 m above sea level in the western portion of the country (*Wallach, 2016*). Therefore, our study increases the number of *Epictia* species currently found in Nicaragua to $n = 2$. As previously mentioned (Comparisons section), *E. rioignis* sp nov. differs from *E. ater* mostly by quantitative characters, such as the number of middorsal scales and snout-vent length as well as qualitative characters such as the presence/absence of a frontal scale, supralabial-supraocular contact, pattern of dorsal

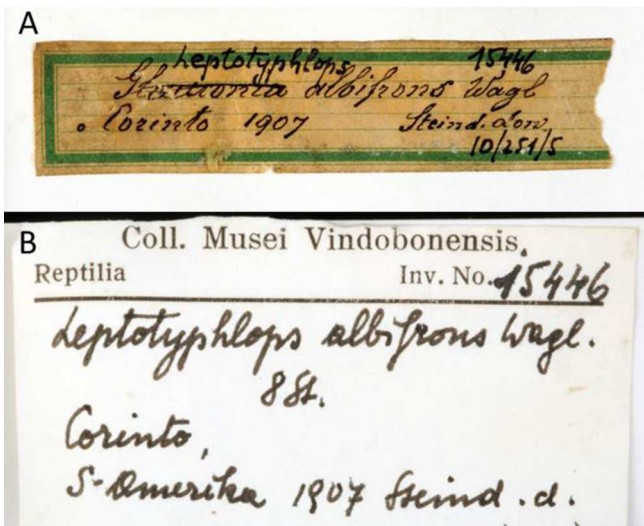

**Figure 16 Handwritten labels of NMW 15446.** (A) By Steindachner and (B) by Eiselt. Photo credits: Alice Schumacher.

stripes on body and tail (present study, (*Wallach, 2016*)), and also based on osteological evidences. Even if there is an apparent overlap of the amplitude of middorsal scales (250–267 in *E. rioignis* sp nov. *versus* 212–266 in *E. ater*), general means for such counts ($\bar{x} = 260.83$ for *E. rioginis* sp nov. *versus* $x = 237.3$ for *E. ater*; present study; (*Wallach, 2016*)) provide evidence for their distinction. Additionally, albeit both supraocular-supralabial contact, as well as the fusion of the frontal and rostral scale can be subject to intraspecific variation (*Wallach, 2016*; *Francisco, Pinto & Fernandes, 2018*), such variation usually occurs in a very limited number of the samples. Thus, these characters are still useful for the differentiation of threadsnake species, especially with respect to the relatively conservative and simplified external morphology of these snakes. The combination of scale size, arrangement, and proportion, color pattern and internal anatomy has proven to be sufficient to distinguish the different taxa, which have also been previously identified in molecular studies (*Wallach, 2016*). Additional studies on species anatomy (such as hemipenial morphology) might reveal further differences between both species. In fact, with respect to the relatively conservative external morphology of threadsnakes (*Wallach, 2016*; *Martins, Passos & Pinto, 2018*), data on hemipenial morphology (and other internal systems) would be very useful to clarify the systematics of threadsnakes.

*Wallach (2016)* recognizes three groups among the *Epictia* species: one from Mesoamerica (*Epictia phenops* species group) and two from South America (*Epictia tesselata* species group and *Epictia albifrons* species group). According to the author, the Mesoamerican assemblage consists of 11 species (*E. ater, E. bakewelli, E. columbi, E. magnamaculata, E. martinezi, E. pauldwyeri, E. phenops, E. resetari, E. schneideri, E. vindumi*, and *E. wynni*). Given the phenotypic similarities and also similar distribution of

*E. rioignis* sp nov., we assign this species to the *E. phenops* species group. Nevertheless, the composition of the three groups still needs to be clarified with further molecular analyses (present study; *Wallach, 2016*).

## CONCLUSIONS

The biodiversity of scolecophidians worldwide is greatly underestimated and often unexpected (*Wallach, 2016*). Although most of the systematic changes in *Epictia* spp. have arisen from molecular studies (or at least provided a start point for additional taxonomical arrangements, see *Wallach, 2016*), morphological studies including both external morphology and internal anatomy are very important for systematic clarification of this group (present study). Even if molecular studies have aided on the identification of cryptic species, this study shows that the morphological analysis of collection specimens still reveals novel data for threadsnakes. This further emphasizes the importance of zoological collections in housing specimens that still allow description of new taxa based on specimens collected more than a century ago.

## ACKNOWLEDGEMENTS

We are grateful to Bettina Riedel, Georg Gassner and Florian Bacher (NMW) for taking tissue samples and helping with investigations on the collecting event and type locality of the new species. Alice Schumacher (NMW) for taking pictures and Josef Muhsil (NMW) for graphic support. For allowing the revision of specimens in their museum collections and/or the loan of material we are indebted to: FL Franco and G Puorto (IBSP), H Zaher (MZUSP), ALC Prudente (MPEG), FF Curcio (UFMT), R Feio (MZUFV), G Colli and M Zatz (CHUNB), M Pires (LZV/ UFOP), C Raxworthy (AMNH), K De Queiroz, R Wilson, R McDiarmid and A Wynn (USNM), C Phillips (UIMNH), JL Watters (OMNH), R Brown (KU), B Hollingsworth (SDNHM), N Camacho (LACM), J Rosado (MCZ), A Resetar (FMNH), S Kretzsmar (FML), J Lynch (ICN), O Torres-Carvajal (QCAZ), P Campbell (BMNH), A Ohler and N Vidal (MNHN), M Franzen (ZSM), F Tillack (ZMB), M Auer (MTKD), L Mogk and G Köhler (SMF), C Aguilar (MUSM), A Scanferla (IBIGEO), R Santa Cruz and H Cárdenas (MUSA), and P Venegas (CORBIDI). Angele Martins is grateful to K De Queiroz, D Johnson, A Nonaka, R McDiarmid and all USNM staff for the support during the conduction of the project in the USA.

## APPENDIX: ADDITIONAL SPECIMENS EXAMINED

*Epictia albifrons*: BRAZIL: **Mato Grosso**: Cuiabá (SMF 16695).
*Epictia albipuncta* (all specimens so far declared as *Leptotyphlops melanotermus*): ARGENTINA: without locality (NMW 16871/1–4); **Cordoba**: San Francisco (MTKD 27438–40); **Tucuman** (FMNH 229948, MCZ 120055, MCZ 120056); BOLIVIA: **Cochabamba**: Alto Chiripiri (FMNH 161503); **La Paz**: La Paz (ZMB 26203).
*Epictia alfredschmidti*: PERU: **Ancash**: Huarmey: Malvas, 2,940–3,090 m a.s.l. (MUSM 20068—Holotype, SMF 80066, SMF 80067—Paratypes).

*Epictia antoniogarciai*: PERU: **Cajamarca**: Jaén: Santa Rosa de la Yunga, 1,268 m a.s.l. (CORBIDI 2069—Paratype, CORBIDI 7678—Holotype); **Amazonas**: Utcubamba: Zapatalgo, 934–968 m a.s.l. (ZFMK 90934—Paratype, ZFMK 96676—Paratype, CORBIDI 5670).

*Epictia ater:* COSTA RICA: **Guanacaste**: Guanacaste NP (ZFMK 57774, ZFMK 57775); NICARAGUA: **Chinandenga:** Volcano Chongo (KU 194336, 2000596); **Estelli:** (KU174119-21, KU174134-35); **Granada**: Volcano Mombacho, 1,100 m a.s.l. (SMF 78981); No data (MNHN 1888.240, MNHN 1888.241); **Managua** (ZSM 660/1911); **Carazo**: 20 km southwest of Diriamba, La Maquina (SMF 81036).

*Epictia australis:* ARGENTINA: **Buenos Aires**: Sierra de la Ventana (IBIGEO 5887–5905); **Cordoba**: Pampa Achala (MTKD 34013); Sierra Chica de Cordoba (ZSM 198/136/1); PARAGUAY: **Chaco**: Chaco (ZFMK 36112), Filadelfia (ZFMK 51301), 35 km west of Filadelfia (ZFMK 53237, ZFMK 53238), Para Todo (ZFMK 55208).

*Epictia collaris:* FRENCH GUYANA: Saül (MNHN 1999.8309); Montagne de Mahury (MNHN 1977.49—Paratype); Camp de Saint Eugène (MNHN 1996.4580).

*Epictia diaplocia*: PERU: Cusco: La Convencion: Echarate: 557 m a.s.l. (MUSM 31322); Pozo Cashiriari 2 (MUSM 26115); Camana: Bosque de Terazas (MUSM 28169); **Huánuco**: Pachitea: Chaglla-Rinconada (CORBIDI 13356, 13357); **Madre de Dios**: 15km E Puerto Maldonado, 200 m a.s.l. (KU 212594); Manu: Manu Learning Center (MUSM 31619); **San Martin**: 14km ESE Shapaja, 360 m a.s.l. (KU 214906); Moyobamba (BMNH 74.8.4.104).

*Epictia fallax:* VENEZUELA: **Vargas**: "Laguayra" [La Guaira] (ZMB 9550—Holotype).

*Epictia goudotii:* COLOMBIA: Valle Magdalena (MNHN 1068—Holotype); VENEZUELA: Aragua (USNM 258148); Bolivar: **Guri** (ZFMK 64415, ZFMK 64416); Distrito Federal: **Tarna** (ZFMK 64423).

*Epictia hobartsmithi:* VENEZUELA: **Bolivar**: Guaiquinima Tepui, 1,180 m a.s.l. (ZSM 107/1990—Paratype).

*Epictia magnamaculata*: HONDURAS: **Islas de la Bahia**: Isla de Utila: Iguana Station (SMF 81215, SMF 79859, SMF 79868, USNM 54760); Isla de Roatan, Mud Hole Bay (SMF 80860); Isla del Cisne Grande, 16 m (SMF 90441); Isla Guanaja: Northeast Bight (SMF 75998).

*Epictia melanura*: PERU: **La Libertad** (FMNH 34269—Holotype); Chiclin (FMNH 34268, Paratype).

*Epictia munoai*: ARGENTINA: **Buenos Aires** (SMF 16694); BRAZIL: **Rio Grande do Sul** (ZMB 1432, ZMB 85279); URUGUAY: Tacuarembo (USNM 163506).

*Epictia peruviana:* PERU: Ayacucho: ca. 132 km from Ayacucho on road to Kirimbi (MUSM 30220); **La Pampa de Sacramento** (ZFMK 41475).

*Epictia phenops:* GUATEMALA: No data (MNHN 5741, MNHN 1667); MEXICO: **Oaxaca**: Tehuantepec (MNHN 8405, MNHN 5660, MNHN 4430); EL SALVADOR: **San Salvador**: San Salvador (SMF 42801, SMF 77236); San Antonio near San Salvador (SMF 42933); Instituto Tropical, 700 m a.s.l. (SMF 42022–42023, 42025, SMF 43216–18, SMF 75814, SMF 77399); km 40 of the Road to San Vicente (SMF 75813); km43 of the Road to San Miguel (SMF 77400); **Sonsonate**: Hacienda San Antonio, 220 m a.s.l. (SMF 42024).

*Epictia rufidorsa*: PERU: Ancash: Recuay: Llacllin: Chaucayan: km 65 Pativilca-Huaraz, 1,320 m a.s.l. (MUSM 25426); **La Libertad**: Chiclin (FMNH 34305); **Lima**: Pv Canta: Campamento Camaná: km 57, Road to Canta (MUSM 3416); Huarochiri: Tornamesa: Santa Cruz de Cocachacia, km 55 Carretera Central (MUSM 25349); Lima (MUSM 16675); Chaclacayo (CORBIDI 6865, MUSM 16733, MUSM 16735); Club Villa Tusan (CORBIDI 10417); Surquillo, San Boja (MUSM 3094); Garden of MUSM (MUSM 10082).

*Epictia septemlineata*: PERU: **Cajamarca**: Celendin: Limon, 2,053 m a.s.l. (CORBIDI 14683—Holotype).

*Epictia signata:* No data (MNHN 3235—Holotype).

*Epictia striatula:* PERU: No data (FMNH 81509); **Cuzco**: Marcapata valley (BMNH 1902.5.29.184, so far declared as *L. melanotermus*); La Convención: CCNN Poyentimari (CORBIDI 8564); **Puno** (FMNH 40210); BOLIVIA: **Chaco** (NMW 15542/4); **Santa Cruz**: Florida: Pampagrande: Pampagrande (ZFMK 66925, ZFMK 66926, ZFMK 66927, ZFMK 75058); Los Negros (ZFMK 75033); Palmasolo (ZFMK 75059).

*Epictia subcrotilla*: PERU: **La Libertad** (FMNH 34304); Chiclin (MCZ 48936); **Piura**: Huancabamba: San Jacinto del Tocto (CORBIDI 14195); Morropon (MUSM 2775); **Cajamarca**: Jaén: Bellavista (CORBIDI 1818).

*Epictia teaguei*: PERU: **Cajamarca**: Rio Zana (FMNH 232568); Chota: Chucrade Chekos (MUSM 28351); La Granja (CORBIDI 1604, MUSA 4339, MUSA 4341, MUSM 28415, MUSM 28416); Querocoto (MUSM 31208); Rio Paltic (MUSM 28414).

*Epictia tenella*: FRENCH GUYANA: No data (MNHN 2001.0319); Haut-Oyapock (MNHN 1978.2583, MNHN 1978.2584); TRINIDAD AND TOBAGO: **Trinidad**: (ZMB 24661); VENEZUELA: **Sucre**: 3–5 km N Macuro (KU 133524).

*Epictia tesselata*: PERU: No data (FMNH 134464); **Ica**: Pisco: Paracas: Punta Arquillo near to Mirador (MUSM 3418); **Lima** (FMNH 35097, FMNH 36726, BMNH 51.1.19.29, MUSM 2820); Callao (MTKD 9304); City of Lima (SMF 41885, MUSM 2383, MUSM 16734—declared as *Epictia teaguei*); Old distrito de cercado building E (KU 212773); Avenida Circunvalacion, Huachipa (CORBIDI 12707); Barranco (MUSM 3092); Jesus Maria, Garden of MUSM (MUSM 24423, MUSM 24424—both declared as *Epictia teaguei*); La Molina, Calle Melgarejo (CORBIDI 12814); Magdalena (MUSM 16123, MUSM 16124); Miraflores (MUSM 14013, MUSM 16673); Monterrico, Las Incas (MUSM 3091); Pueblo Libre (MUSM 3179); San Luis, Tupac Amaru (MUSM 3171); Santa Anita (MUSM 25378); University San Marcos (MUSM 3374).

*Epictia tricolor*: PERU: **Ancash**: Carhuaz, 2,710 m a.s.l. (MUSA 2873); Huaraz, 3,250 m a.s.l. (KU 135176, KU 135177); Huaylas: Yuanca Pampa, 2,700 m a.s.l. (MUSM 669—Holotype, MUSM 670—Paratype, MUSM 2923, USNM 195853); **Cajamarca**: Cajabamba, 2,700 m a.s.l. (KU 135178), 2,880 m a.s.l. (MUSA 4234, MUSA 4235, MUSA 4236, MUSA 4254); **La Libertad**: Sanchez Cavion, Sanagoran (MUSM 28550).

*Epictia vanwallachi*: PERU: **La Libertad**: Pataz: Vijus, 1,297 m a.s.l. (CORBIDI 14682—Holotype).

*Epictia vonmayi*: PERU: **Cajamarca**: Chota: Querecoto: La Granja-Río Tinto, 2,069 m a.s.l. (MUSA 4342—Holotype), 1,985 m a.s.l. (MUSA 4340—Paratype).

*Epictia vellardi:* PARAGUAY: **Boquerón**: Toledo (MTKD 43465–43467, 43469).

*Epictia venegasi*: PERU: **Cajamarca**: Cajabamba: Cachi-Moyan, 2,551 m a.s.l. (MUSA 4252—Holotype), 2,567–2,736 m a.s.l. (MUSA 4237–42, MUSA 4248, MUSA 4253—Paratypes).

*Mitophis asbolepis*: DOMINICAN REPUBLIC: **Barahona**: Loma del Aguacate (USNM 236660).

*Mitophis calypso*: DOMINICAN REPUBLIC: **Samana**: Las Galeras (USNM 236658, 236659).

*Mitophis leptepileptus*: HAITI: **Sud-Est:** Soliette (KU 275542, 275543, 275548, 275549, 275562, 275567, USNM 576217); Fond Verettes (USNM 236661).

*Mitophis pyrites:* DOMINICAN REPUBLIC: **Pedernales**: Pedernales (MCZ 77239, USNM 152452).

*Rena dulcis*: UNITED STATES OF AMERICA: **Florida** (USNM 15466); **Oklahoma**: Comanche (OMNH 35564, 35580); Marshall (USNM 258213, 258214, 258217, 258218); Seminole (USNM 198022, 198023, USNM 258197); **Texas**: Bell (USNM 198036); Frio (LACM 162021); Hays (AMNH 160152, LACM 162022); Hidalgo (USNM 299630, 299642); McLennan Co (FMNH 40957, USNM 198037, 198042, 258186, 258188); Pecos (USNM 7296); Travis Austin (USNM 161288).

*Rena humilis*: MEXICO: **Baja California** (AMNH 5576); **Jalisco** (AMNH 94866); **Nayarit** (AMNH 75587, 87584, 96616, USNM 240693, 240879); **Sinaloa** (AMNH 90761, 90762); **Sonora**: Bahia Kino (USNM 214128); **without locality** (AMNH 66170, USNM 26140); UNITED STATES OF AMERICA: **Arizona:** Graham (AMNH 63458); Maricopa (USNM 246639); Pima (AMNH 73716); Yuma (USNM 26289); **California**: Imperial (USNM 139916, 139917, 222794); Riverside (AMNH 84309); San Bernardino (AMNH 131165); San Diego (FMNH 77599, SDNHM 2956, 9347, 24273, 24407, 24408, 33950, 34302, 42772, 61052, 61055, 61057, 61059, 61060, 61062, 61068, 61088, USNM 193023, 196570, UIMNH 84591).

*Rena maxima* ($n = 1$). MEXICO: **Guerrero:** Chilpancinga (MCZ 33606).

*Rena myopica*: UNITED STATES OF AMERICA: **New mexico:** Eddy: Malaga (USNM 125131); MEXICO: **San Luis Potosí** (AMNH 172551); **Tamaulipas:** Gomez Farias (USNM 248499).

*Rena segrega:* MEXICO: **Coahuila:** Torreon (USNM 193593); UNITED STATES OF AMERICA: **ARIZONA:** without locality (AMNH 43439); Pima (USNM 16952); **New Mexico:** Doña Ana (AMNH 172545,172546, LACM 2154, 134007, 134009); **Texas:** Val Verde (AMNH 172548); Brewster (USNM 103670); **without locality** (AMNH 112249).

*Rena unguirostris*: ARGENTINA: Poman Puesto Rio Blanco (FML 1399); **Tinogasta**: Palo Blanco (FML 1773).

*Siagonodon cupinensis:* BRAZIL: **Mato Grosso:** Barra do Tapirapés: Guarantã do Norte (UFMT 5627).

*Siagonodon septemstriatus:* SURINAME: without locality (ZSM 127/1947, ZMB 3876—Holotype); BRAZIL: **Amazonas**: Manaus: Rio Negro: Estuary of Rio Negro (NMW 15476/1).

*Tetracheilostoma billineatum*: MARTINIQUE: **Le Lamentin:** without locality (USNM 564808, 564809); SAINT LUCIA: **Saint Lucia:** Anse-La- Raye (USNM 222954); **without locality** (MCZ 10693).

*Tetracheilostoma breuili*: SAINT LUCIA: **Islas Maria:** Isla Maria Major (USNM 564810–17).

*Tetracheilostoma carlae:* BARBADOS: **Saint Joseph:** Bonwell (USNM 564818–19).

*Tricheilostoma bicolor*: GHANA: **Somanya:** Krobo (MCZ 55388, MCZ 55383).

*Tricheilostoma sundewalli*: GHANA: **Somanya:** Krobo (MCZ 55396).

*Trilepida brasiliensis:* BRAZIL: **Mato Grosso Do Sul:** Corumbá (UFMT 683, 1159, 1160, 1162, 1163, 1169); Rosário d'Oeste (MNRJ 24334).

*Trilepida dimidiata*: BRAZIL: **Roraina**: Ilha de Maracara: Maracara (BMNH 1994.241).

*Trilepida fuliginosa*: BRAZIL: **Goiás:** Colinas do Sul (MNRJ 19223); Luziânia (CHUNB 40847, 408486); Minaçú (MZUSP 11019); Ouvidor (MNRJ 19221); **Minas Gerais:** Unaí (MNRJ 24400).

*Trilepida jani*: BRAZIL: **Minas Gerais:** Ouro Preto (LZV 813S); Ouro Branco (778S1); without locality (MNRJ 16990).

*Trilepida joshuai*: COLOMBIA: **Antioquia**: Jericó (IBSP 8919); **Valle del Cauca**: Tocota, 1,800 m a.s.l. (NMW 38424/1–2, declared as *Epictia weyrauchi*).

*Trilepida koppesi*: BRAZIL: **Goiás:** Caldas Novas (MZUSP 11111); Aporé (MNRJ 24715, 24716); Luiziânia (CHUNB 40788); Mineiros (CHUNB 25714); **Tocantins:** Porto Alegre do Tocantins (CHUNB 38928).

*Trilepida macrolepis*: BRAZIL: **Pará:** Paraupebas: Floresta Nacional de Carajás (MPEG 23017); COLOMBIA: **Vale Del Cauca**: Buenaventura (USNM 154031, 267261); **Córdoba:** Pueblo Nuevo (ICN 7677); ECUADOR: **Esmeraldas:** Durango (QCAZ 12494); VENEZUELA: **Merida** (NMW 15468/1–4); **Yaracuy:** Salom (MTKD 26320–26321).

*Trilepida salgueiroi*: BRAZIL: **Bahia**: São José do Macuco (currently São José da Vitória) (MZUSP 9098); **Espírito Santo:** Governador Lindemberg (MNRJ 12132); Itá (currently Baixo Guandu) (IB 8876, holotype); Aracruz (MNRJ 4856); Campinho (MNRJ 1925); Governador Lindemberg (MNRJ 12131-12132); Linhares, Goytacazes (MNRJ 1926); Linhares, Sooretama (MZUSP 2463); **Minas Gerais:** Aimorés (MCNR 1468-1469, MNRJ 12239), Muriaé: (MZUFV 1519); Recreio (MNRJ 7856); **Rio De Janeiro:** Cambuci (MNRJ 14487); Niterói: Itaipu (MNRJ 13124,15422).

*Trilepida pastusa*: ECUADOR: **Carchi**: Tulcán (QCAZ 5778).

### Funding

Angele Martins was supported by the Short-Term Fellowship Grant of the Smithsonian Institute, by a scholarship (process 99999.010032/2014-02) of the Coordenação de Aperfeiçoamento de Pessoal de Nível Superior (CAPES) provided during her PhD, and by Fundação Carlos Chagas Filho de Amparo à Pesquisa no Estado do Rio de Janeiro (FAPERJ, E-26/202.403/2017). The funders had no role in study design, data collection and analysis, decision to publish, or preparation of the manuscript.

## Grant Disclosures

The following grant information was disclosed by the authors:

Short-Term Fellowship Grant of the Smithsonian Institute.

Coordenação de Aperfeiçoamento de Pessoal de Nível Superior (CAPES).

Fundação Carlos Chagas Filho de Amparo à Pesquisa no Estado do Rio de Janeiro: (FAPERJ, E-26/202.403/2017).

## Competing Interests

The authors declare there are no competing interests.

## Author Contributions

- Claudia Koch conceived and designed the experiments, performed the experiments, analyzed the data, contributed reagents/materials/analysis tools, prepared figures and/or tables, authored or reviewed drafts of the paper, approved the final draft.
- Angele Martins and Silke Schweiger conceived and designed the experiments, performed the experiments, analyzed the data, contributed reagents/materials/analysis tools, prepared figures and/or tables, authored or reviewed drafts of the paper.

## Data Availability

The specimens of the new species are deposited in the herpetological collection of the Natural History Museum Vienna (NMW), voucher numbers are: NMW 15446:1–8.

The raw data of the micro-CT scans is available at MorphDBase:

Koch, C. (2019): C_Koch_20190502-M-6.1.

Direct link: http://www.morphdbase.de/?C_Koch_20190502-M-6.1.

Koch, C. (2019): C_Koch_20190502-M-7.1.

Direct link: http://www.morphdbase.de/?C_Koch_20190502-M-7.1.

Koch, C. (2019): C_Koch_20190502-M-8.1.

Direct link: http://www.morphdbase.de/?C_Koch_20190502-M-8.1.

Koch, C. (2019): C_Koch_20190502-M-9.1.

Direct link: http://www.morphdbase.de/?C_Koch_20190502-M-9.1.

Koch, C. (2019): C_Koch_20190502-M-10.1.

Direct link: http://www.morphdbase.de/?C_Koch_20190502-M-10.1.

## New Species Registration

The following information was supplied regarding the registration of a newly described species:

Publication LSID:

urn:lsid:zoobank.org:pub:C2392EA5-9957-45BF-AE8C-0998E342F90B.

*Epictia rioignis* sp. nov. LSID:

urn:lsid:zoobank.org:act:4C7CFC14-FDC7-4B41-B7CF-7CCB8F0E8649.

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
