# Peer review of "A century of waiting: description of a new Epictia Gray, 1845 (Serpentes: Leptotyphlopidae) based on specimens housed for more than 100 years in the collection of the Natural History Museum Vienna (NMW)"

_PeerJ, doi:10.7717/peerj.7411_

## Round 0.1 · original submission · Minor Revisions

Dear authors

Your ms has now been reviewed and found interesting. As you can see, our reviewers suggest a number of changes, which should not be too difficult.

Be aware that we can send your revision to the same reviewers.

Kind regards
Michael Wink
AE

·

Basic reporting

The manuscript is well written and easy to follow. Main data are reported in a rather clear way and supported with figures of good quality. As far as I can see, all the most significant features are clearly depicted in the figures, thus allowing both a better comprehension of them and the possibility for the reader to easily verify them by looking at the original specimens.

Experimental design

I have no significant issues to highlight regarding how the whole study was set up. Indeed, the authors follow a standard taxonomical approach and provide all the most important information about how they conducted their study. Particularly remarkable is the large amount of specimens of other species (in particular Epictia species) that are used as comparative material: this is very important in works in which new species are described, given that using a little comparative sample might lead to an underestimation of the variability of the already known taxa. All the comparative specimens used are listed as an appendix, which is also good.

Validity of the findings

Even though I have to admit I am not a specialist of South American snakes, the results reported by the authors seems really convincing to me and their conclusions appear based on sound data. Combining external morphological characters with osteological features is really appreciable. Skeletal anatomy of scolecophidian snakes is rather poorly known, particularly as far as the skull is concerned. New osteological data for these reptiles are therefore really welcome, also because they will allow future researchers to compare extant taxa with fossils attributed to scolecophidians (a thing that would not be possible otherwise).
One thing I would like to ask the authors to add in their manuscript, if possible, is some brief comments on the general morphology of the trunk vertebrae when discussing the variation. Trunk vertebrae make up most of the known fossil record of scolecophidian snakes, but they are generally considered to be almost non-significant from a taxonomical point of view below Scolecophidia as a whole. I definitely understand that a description of the vertebral morphology is completely out of the scope of this paper and it would be too much, of course. However, could it be possible, based on the raw data already available, to state whether vertebrae of these snakes (even considering the Epictia genus alone, but also compared with other genera in case) do show some kind of difference that might have a taxonomical significance (even if subtle and only in combination with other features)?

Additional comments

A list of other, minor comments follows:

Line 275: The plural of lamina should be laminae.

Line 279: “pierces” instead of “pierce”

Line 340: Here the “Pelvic girdle” heading should have the same rank as “Suspensorium”, “Cervical vertebrae”, ecc., I guess.

Lines 388-389: Is “first precloacal” correct? Or is this the first cloacal?

Lines 388-391: When reporting the position of the pelvic girdle, can you please clarify the distance in terms of number of vertebrae of the anteriormost trunk vertebra to which the girdle is related compared to the cloacal region, if possible? For example: “Pelvic girdle located at the level of the 230th (penultimate precloacal) and first cloacal vertebrae (Specimen X)”.

Line 395: A semicolon is missing after “(n=1)”.

Line 397: Is it the femur (as stated here) or the pubis (as stated previously in the description) that is fused to the ischium in the holotype?

Figure 13: In Fig. 13B, labels indicating the different parts composing the girdle would be useful for readers not familiar with snake postcranial osteology to better interpret the figure.

Reviewer 2 ·

Basic reporting

See below my "General comments"

Experimental design

See below my "General comments"

Validity of the findings

See below my "General comments"

Additional comments

This paper provides a detailed documentation for a new scolecophidian snake, a reptile group that is still insufficiently known in many aspects. I agree with the authors on the species level distinction of this taxon and the establishment, therefore, of a new species. The rather detailed descriptions will be rather helpful for future comparisons with other taxa. The quality of the photographs is high. Especially the documentation of the skeletal anatomy appears rather important. The language, grammar, and the structure of the text are well presented.
I have to note however, that there is important “unbalance” between the description of cranial and postcranial elements. Of course the skull has the most important characters but vertebrae in snakes have so specialized anatomy and moreover, they are in fact the ONLY elements (so far) consisting their fossil record.
So, the main issue I would like to raise here is the insufficient description of the vertebral anatomy and the fact that the authors figure only the atlas and axis (and only in dorsal and lateral view; Fig. 10) and few vertebrae that appear in Fig. 13 (only in lateral view). Furthermore, their description is confined exclusively on the cervical area mentioning only the presence of hypapophyses, which is a rather basic information only. In a recent paper, Georgalis et al. (2017) addressed the currently poor knowledge of the vertebral anatomy of scolecophidians which as a result does not allow determination of fossil specimens of this group, not even for identifying them to the family level (typhlopids or leptotyphlopids). Considering that trunk vertebrae (especially mid-trunk ones) are the most common finds of scolecophidians in the fossil record, it is here a good opportunity to Figure and describe the vertebral anatomy of a trunk vertebra. What I mean precisely is that, as the authors have already these CT scans of the holotype, it is a gold opportunity to isolate one trunk vertebra (ideally one mid-trunk) and figure it in all standard five views (anterior, posterior, dorsal, ventral, lateral) with the proper orientation. For a proper orientation of a scolecophidian vertebra and also a description, please see Georgalis et al. (2017:fig. 5). In that way, important structures of the vertebra, such as the zygosphene, zygantrum, neural arch, cotyle, condyle, prezygapophyses, etc.. will be visible. In their current form, these Figures unfortunately provide no clue about these structures, which are nevertheless crucial for snake taxonomy (e.g., extent of prezygapophyses, concavity and thickness of the zygosphene, vaultness of the neural arch etc.). This will eventually provide important information and insight. Ideally, this could be done also for a caudal vertebra, but this is not so so necessary.
Georgalis, G.L., Villa, A., and Delfino, M., 2017a. Fossil lizards and snakes from Ano Metochi – a diverse squamate fauna from the latest Miocene of northern Greece. Historical Biology 29:730–742.

Few other comments on the text:
- In the Introduction it would be good to mention that their vertebral anatomy is so insufficiently known. This has implications for their not proper recognizing in the fossil record and the fact there are major difficulties in assigning fossil specimens to a particular family, leaving them almost always as Scolecophidia indet. (see Georgalis et al. 2017). Afterall you describe their skeletal anatomy of your new species, so it would be interesting to highlight this.
-Diagnosis part: Is there any distinctive feature in the vertebrae (apart from their total number already mentioned) that could be used in the Diagnosis? Of course the vertebral anatomy of scolecophidians is so insufficiently known, but in case the authors distinguish something, it would be interesting to add it in the diagnosis.
Line 69: The correct abbreviation for Natural History Museum in London is NHMUK
Line 169 and afterwards: Every taxon that appears for the first time in the text should not have its genus abbreviated. So all these Epictia species mentioned here, in their first mentions should have their full generic names writen and not just “E.”

Overall, this is an important contribution and will augment our knowledge on the diversity, external morphology and cranial anatomy of leptotyphlopids. Hopefully also on their vertebral anatomy
I am at your disposal for any further querry

·

Basic reporting

Comments on the original document.

Experimental design

No comment

Validity of the findings

Comments on the original document.

Additional comments

The article describe a new species of Epictia based on eight specimens from "Corinto, S. America", that was housed in the collection of the Natural History Museum Vienna for more than a century. Authors include large comparisons within congeneric Epictini, and proved relevant argumments on descriptions. They also proved important argumments to restrict type locality to Corinto, Nigaragua. Beside some minor revisions, I recommend this article for publishing in PeerJ.

---

## Round 0.2 · accepted · Accept

Dear authors

Thanks for your revision which is adequate. Therefore, we can now accept your ms.

Best regards
Michael Wink